# Fluid pressurisation and earthquake propagation in the Hikurangi subduction zone

S. Aretusini [1✉], F. Meneghini[2], E. Spagnuolo [1], C. W. Harbord[3] & G. Di Toro [1,4]

In subduction zones, seismic slip at shallow crustal depths can lead to the generation of tsunamis. Large slip displacements during tsunamogenic earthquakes are attributed to the low coseismic shear strength of the fluid-saturated and non-lithified clay-rich fault rocks. However, because of experimental challenges in confining these materials, the physical processes responsible for the coseismic reduction in fault shear strength are poorly understood. Using a novel experimental setup, we measured pore fluid pressure during simulated seismic slip in clay-rich materials sampled from the deep oceanic drilling of the Pāpaku thrust (Hikurangi subduction zone, New Zealand). Here, we show that at seismic velocity, shear-induced dilatancy is followed by pressurisation of fluids. The thermal and mechanical pressurisation of fluids, enhanced by the low permeability of the fault, reduces the energy required to propagate earthquake rupture. We suggest that fluid-saturated clay-rich sediments, occurring at shallow depth in subduction zones, can promote earthquake rupture propagation and slip because of their low permeability and tendency to pressurise when sheared at seismic slip velocities.

[1] HPHT Laboratory, INGV, Rome, Italy. [2] Department of Earth Sciences, University of Pisa, Pisa, Italy. [3] Department of Earth Sciences, University College London, London, UK. [4] Dipartimento di Geoscienze, University of Padua, Padua, Italy. ✉email: stefano.aretusini@ingv.it

Earthquakes that propagate along the plate interface in subduction zones at shallow crustal depths can generate tsunamis, a significant natural hazard in numerous countries around the Pacific and Indian oceans[1]. The Hikurangi subduction zone offshore New Zealand hosted two moment magnitude (Mw) 7.0–7.2 tsunami earthquakes in 1947[2,3]. In the same area, there is evidence of slow slip events (SSEs) propagating to within 2 km of the seafloor (SSEs)[4–6], indicating that the very shallow plate boundary megathrust at Hikurangi may host both large earthquakes and aseismic slow slip, which may propagate all the way to the trench. In many cases, SSEs precede large subduction zone earthquakes (e.g. Tohoku 2011 Mw 9.0[7], Iquique 2014 Mw 8.1[8]). Thus, there is a growing concern, both within the Hikurangi subduction zone and worldwide, regarding earthquakes that could propagate to shallow depths and result in the generation of a tsunami.

In 2019, the International Ocean Discovery Programme (IODP) Expedition 375 recovered fluid-saturated clay-rich fault zone materials from the Pāpaku thrust, sited within the zone of SSEs and historical seismicity in Hikurangi subduction zone[9]. Scientific drilling of this area represents a unique opportunity to study the mechanics of earthquake rupture within an active tsunamigenic fault.

Theoretical studies suggest that thermal pressurisation of pore fluids trapped in fault materials can reduce the dynamic shear strength of faults during seismic slip[10–12]. In low permeability and velocity strengthening, clay-rich materials typical of subduction forearcs, dynamic weakening behaviour at seismic sliding velocity occurs over a short distance so that a negligible mechanical work is dissipated by the seismic rupture[13,14]. The combination of low dynamic fault strength and short weakening distances enables rupture propagation in shallow sections of the fault and also promotes large seismic slip[15]. However, laboratory experiments designed to test theoretical models of coseismic fluid pressurisation have been limited by the technical challenge of imposing realistic normal stress and pore fluid pressure on non-lithified fault materials sheared at seismic deformation conditions. For instance, fluids and non-lithified materials must be sealed and confined, respectively, to avoid extrusion of the sample under application of normal stress at imposed slip velocities of ~1 m/s, typical of crustal earthquakes[16].

Here, by exploiting a new experimental set-up, we shear Pāpaku thrust clay-rich fault materials at seismic slip velocities under fluid-pressurised conditions. Here, we show that Pāpaku thrust materials sustain high shear stress at the onset of slip, which dynamically weakens to low shear stress as a direct result of pore fluid pressure changes. After coseismic shear-induced dilatant strengthening, Pāpaku thrust fault materials display pressurisation of pore fluid resulting in dynamic weakening behaviour and low breakdown work, which could allow rupture propagation and promote large seismic slip in the shallow section of the subduction zone.

## Results and discussion
**Pāpaku thrust**. The IODP expeditions 372 and 375 drilled, logged, and cored the Hikurangi subduction zone, offshore of the North Island, New Zealand. In this area, the Pāpaku thrust is a shallow branch of the plate boundary fault, which has hosted historic tsunami earthquakes and more recently shallow SSEs (Fig. 1a, b). Drilling and sampling of the Pāpaku thrust fault rocks occurred at site U1518 (Fig. 1b), down to 490 m below seafloor (mbsf). The thrust is defined by a 55-m-thick fault zone (305–360 mbsf), including a principal (305–325 mbsf) and a secondary (350–360 mbsf) fault core. The top of Pāpaku thrust fault is characterised by a ~0.5 Ma age inversion[17].

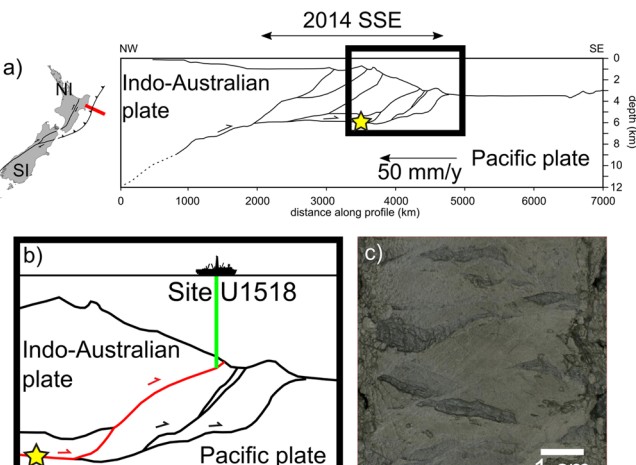

**Fig. 1 Geological setting of Pāpaku thrust and fault core materials. a** Transect across Hikurangi subduction zone (red segment, Gisborne, Northern Island, New Zealand), redrawn from interpreted seismic profile[52]. Here the Pacific Plate is subducting below the Indo-Australian plate with a convergence rate of 50 mm/year[4]. In this area, a tsunami earthquake occurred in 1947 (yellow star in **b** is the estimated hypocentral depth) and an SSE in 2014 (black arrows). **b** In 2019, the IODP expedition 375 drilled the upper plate in site U1518 (green line), down to ~490 mbsf, intercepting the Pāpaku thrust at ca. 300 mbsf (highlighted in red). **c** Pāpaku thrust principal fault core materials are enriched in clay minerals (scan of core 14R1A where the fault core materials were sampled from).

We selected rock materials deriving from three intervals with respect to the fault zone: (i) the hanging wall (~9 m above the fault zone), (ii) the principal fault core, and (iii) the footwall (~19 m below the fault zone). The three samples are clay-rich sediments with an average mineral composition of $45.4 \pm 2.1$ wt % total clay minerals (smectite + illite + chlorite + kaolinite), $28.7 \pm 0.8$ wt% quartz, $17.2 \pm 0.6$ wt% feldspars, and $10.8 \pm 0.8$ wt % calcite[18]. Fault core materials (Fig. 1c) have lower smectite content compared to the footwall and hanging wall materials (i.e. 14.2 versus 22.3 and 21 wt%, respectively).

**Permeability and mechanical behaviour of Pāpaku thrust materials**. We measured the permeability of remoulded Pāpaku thrust materials under hydrostatic conditions in a permeameter (Supplementary Figure 1). Measurements were performed at effective stresses ranging from 3 to 100 MPa using the pore fluid oscillation technique (see "Methods" and Supplementary Table 1). At 3 MPa effective normal stress, the in situ conditions of the Pāpaku thrust equivalent to ca. 300 m depth and hydrostatic pore fluid pressure condition, permeability is lower in the hanging wall ($1.66 \pm 0.02 \times 10^{-20}$ m$^2$) compared to the principal fault core ($3.95 \pm 0.19 \times 10^{-20}$ m$^2$) and to the footwall ($7.25 \pm 0.82 \times 10^{-20}$ m$^2$) materials.

We performed 11 experiments with a high-velocity friction machine, capable of imposing slip velocities ranging from $10^{-5}$ to 5.35 m/s, during which we measure the dynamic evolution of the shear strength (see "Methods"). The evolution of shear strength during an experiment is considered as a guide to the evolution of material strength during an earthquake. In addition to the measurements of shear strength, normal stress, axial shortening, displacement, and velocity[13,14,19], we also measured (1) the pore fluid pressure with transducers located 3 and 70 mm upstream and downstream of the gouge layer, respectively, and (2) temperature with a thermocouple located on the upstream gouge layer boundary. This represents a novel experimental set-up allowing the pressurisation of fluids in fault gouge samples (for

description and calibration of the sample assemblage, see "Methods" and Supplementary Figure 4). The temperature evolution across the gouge layer was estimated with numerical methods (i.e. 1D finite-difference model, backward Euler method), accounting for the measured temperature at the edge of the gouge layer (see "Methods"). Six experiments were performed on remoulded samples derived from the fault core, hanging wall and footwall of the Pāpaku thrust. We also performed an additional experiment using higher permeability Carrara marble gouge, which is also a well-studied standard selected to be used as a benchmark. All seven experiments were performed at room temperature (ca. 25 °C), at a normal stress of 6 MPa, pore fluid pressure of 3 MPa, and confining pressure of 4 MPa. Pore fluid pressure on the upstream side of the sample was controlled at a constant value (3 MPa), whereas on the downstream side pore fluid pressure was undrained to measure spontaneous changes during deformation. Four additional experiments were performed on remoulded fault core materials under room humidity and water-dampened conditions, at room temperature (25 °C), and a normal stress of 3 MPa so that the effective normal stress was identical to the fluid-pressurised experiments. In all of the experiments, one low-velocity slip pulse was performed at $10^{-5}$ m/s for 0.01 m displacement to achieve gouge compaction. Then, we unloaded the shear stress from the experimental fault. Afterwards, a seismic velocity slip pulse up to 0.8 m/s for 0.4 m of slip was imposed to simulate moderate (Mw 6.0–6.5) tsunamigenic earthquakes or the early stages of larger magnitude earthquakes.

In the seismic velocity slip pulse, during initial acceleration to 0.8 m/s, shear strength increased nearly instantaneously to a peak value $\tau_p$ resulting from elastic loading of the sample (Fig. 2a, c). Afterwards, shear strength decreased to a residual value $\tau_r$ ~75% lower than $\tau_p$ (Fig. 2a). This decrease of shear strength, typically occurring during seismic slip, is defined as dynamic weakening[10,20]. The shear strength decay with displacement

was fitted with a power-law function to define $\tau_p$, and $\tau_r$, the slip weakening distance $D_w$ and the exponent of the power law $\alpha$ (see "Methods" and Supplementary Figure 5).

Fault core materials showed the lowest peak strength, residual strength, and slip weakening distance under fluid-pressurised conditions than under water-dampened and room humidity conditions (Supplementary Figure 5a–d). In fluid-pressurised experiments, fault core materials had a shorter slip weakening distance with respect to the hanging wall and footwall materials, but all materials displayed similar peak and residual strength (Supplementary Figure 5e–h).

In fluid-pressurised experiments performed on Pāpaku thrust materials, the pore fluid pressure measured upstream ($P_{f,u}$) and downstream ($P_{f,d}$) fluctuated during the experiment. During initial slip acceleration to 0.8 m/s, $P_{f,d}$ decreased below the imposed 3 MPa, and increased either during or after the seismic velocity slip pulse (Fig. 2). A drop in downstream pore fluid pressure of 0.51 ± 0.11 MPa occurring ca. 0.08 s after the slip initiation was associated to gouge layer thickness increase of 6 ± 4 μm (Fig. 2a and Supplementary Table 2). Pore pressure drop and increase of gouge thickness are consistent with shear-induced dilatancy at slip initiation. Independently of the origin of the sheared gouges (footwall, hanging wall, and fault core), the maximum downstream increase $\Delta P_f$ ($=P_{f,d} - P_{f,u}$) was 0.82 ± 0.28 MPa and was measured 43.16 ± 24.46 s after the initiation of the slip pulse (Fig. 2b and Supplementary Table 2). A gouge layer thickness reduction of 132 ± 72 μm occurred during the seismic slip (Supplementary Table 2). The fluid-pressurised experiments performed on Carrara marble gouge under identical loading conditions aided interpretation of the experiments performed with the less permeable Pāpaku thrust materials (Fig. 2c). In the sheared marble gouges, after the initial peak in strength at slip initiation, a drop in $P_f$ (ca. 1.87 MPa in 0.08 s) coincided with an expansion of gouge layer thickness (ca. 69 μm) and with a second peak in strength that interrupted the dynamic weakening. This

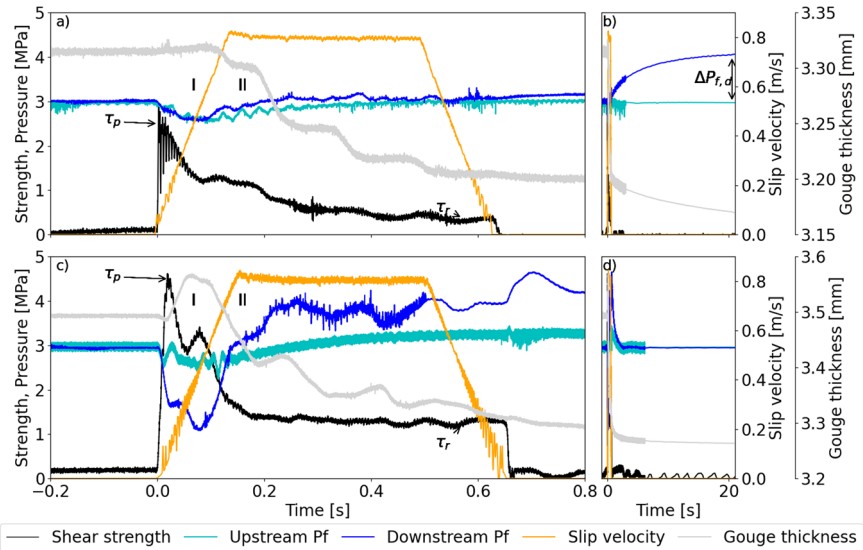

**Fig. 2 Seismic mechanical behaviour of pressurised fault gouges.** Anatomy of the seismic velocity slip pulse in Pāpaku thrust footwall material (experiment s1737) and Carrara marble gouge (experiment s1823). Slip velocity, thickness, shear strength and pore fluid pressure measured on the downstream and upstream side are presented versus time. Pāpaku thrust materials. **a** At the onset of slip (I), slip velocity increases to the target value of 0.8 m/s, pore fluid pressure decreases right after the achievement of the peak strength ($\tau_p$). Afterwards (II), shear strength decreases to the residual value ($\tau_r$), while pore fluid pressure increases (faster downstream than upstream). **b** At the end of slip pulse, pore fluid pressure gradually increases in the downstream side until it achieves a maximum value ($\Delta P_f$). Carrara marble gouge. **c** At the onset of slip (I), slip velocity increases to the target value of 0.8 m/s, pore fluid pressure decreases during the achievement of the peak strength ($\tau_p$). Afterwards (II), shear strength decreases to the residual value ($\tau_r$), while pore fluid pressure increases (faster downstream than upstream). **d** At the end of the slip pulse, pore fluid pressure is equilibrated in the whole system after ca. 5 s.

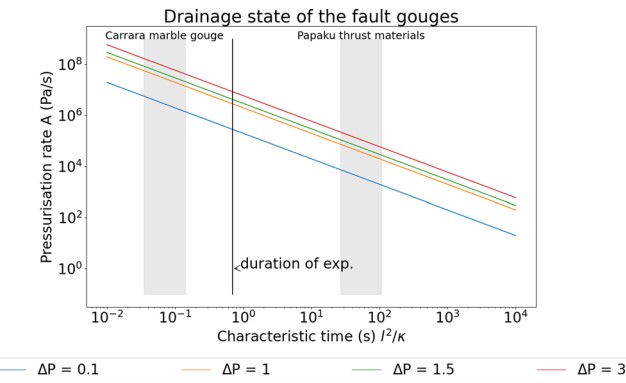

**Fig. 3 Characteristic times of diffusion in pressurised fault gouges.** In Carrara marble gouge the characteristic time of diffusion (see "Methods," grey shaded area on the left) was lower than the duration of the seismic slip pulse indicating drained deformation. On the other hand, in Pāpaku thrust materials (grey shaded area on the right), the characteristic time of diffusion was larger than the duration of the seismic slip pulse, indicating undrained deformation. Under constant pressurisation rate $A$, the magnitude of the pore fluid overpressure $\Delta P$ decreases with the increase of characteristic times of diffusion: the more permeable Carrara marble gouge is less likely to pressurise than the less permeable Pāpaku thrust materials. Pore fluid overpressures of 0.1 (blue line), 1 (orange line), 1.5 (green line), and 3 (red line) MPa are indicated in the plot[21].

second peak was followed by a progressive increase of $P_{f,d}$ (up to 1 MPa) occurring during the decrease in shear strength. A gouge layer thickness reduction of ca. 186 μm occurred during the experiment. Carrara marble gouge had a characteristic diffusion time (0.03 s) three orders of magnitude smaller than Pāpaku thrust materials (i.e. 26.9 s) and 20 times shorter than the duration of the seismic slip pulse (ca. 0.63 s, see Fig. 3). Therefore, Carrara marble gouge was drained during the slip pulse and the pressure rise measured by the downstream pressure transducer was detected almost instantaneously to when it occurred in the slipping zone. Conversely, due to their lower permeability, the pressure increase of the clay-rich gouges of the Pāpaku thrust was detected by the same transducer several tens of seconds after the end of slip[21].

Shear-induced dilatancy was shown to have a stabilising effect and compaction to have a destabilising effect on faults during the earthquake nucleation stage in experiments performed on fluid-pressurised quartz gouges[22,23]. At coseismic slip rates, dilatancy was recently demonstrated to occur during failure of intact rock, a process that may suppress thermal pressurisation during the dynamic weakening of faults[24]. However, the role of dilatancy and compaction in controlling fault seismic slip was never observed, to our knowledge, in non-lithified fault gouges. The competition between dilatancy and compaction was only suggested to be active at seismic slip rates in experiments performed on fluid-pressurised calcite gouges[25,26]. Our experimental configuration was designed to address the experimental limitations discussed in Rempe et al.[26], in particular: we reduced the distance of the downstream pressure transducer from the gouge layer (>184 mm in Rempe et al.[26], 70 mm here) and minimised the volume of the downstream reservoir (>25.5 mL in Rempe et al.[26], 4.7 mL here). These implementations improved the measurement of the short-lived pressure transients occurring in the gouge layer and the complex evolution of pore fluid pressure during fault slip. Shear-induced dilatancy is observed during initial slip acceleration, manifested by a gouge layer increase and fluid pressure drop. With increasing slip, pore fluid pressure increases after ca. 0.1 s, which is accompanied by the onset of thickness decrease indicating compaction of the gouge

layer, which results in a pore pressure increase (Fig. 2a, c). Pore fluid pressure increase during seismic slip can occur by a combination of mechanisms that concur in decreasing the shear strength of the fault by decreasing the effective normal stress. These mechanisms include (1) thermal pressurisation[10]: frictional heating during seismic slip results in a pressure increase in the slipping zone since the thermal expansion of pore fluid is higher than that of the solid matrix; (2) dehydration[27,28]: the temperature increase due to frictional heating is high enough (i.e. $T > 90\,°C$) to dissociate water from the clay minerals; (3) mechanical pressurisation[21]: the pore fluid pressurises because the pore volume decreases by compaction during shear. In our pore fluid-pressurised experiments, dynamic weakening was associated to pore fluid pressure increase (Fig. 2) and the estimated maximum temperature in the centre of the slipping zone was $83.8 \pm 15.0\,°C$ (Supplementary Table 2 and Supplementary Figure 7). The latter suggests that a combination of thermal pressurisation, dehydration, and shear compaction drove the pore fluid pressure increase and therefore the shear strength reduction. In the water-dampened experiments, the estimated maximum temperature of $161.8 \pm 12.5\,°C$ (Supplementary Table 2 and Supplementary Figure 8) suggested that similar processes occurred. On the other hand, in the room humidity experiments, the estimated maximum temperature of $301.5 \pm 5.8\,°C$ (Supplementary Table 2 and Supplementary Figure 8) and the absence of water suggests that dehydration contributes to dynamic weakening.

**Propagation of seismic slip at shallow crustal depths.** The breakdown work $W_b$ defines the energy dissipated during the loss of fault strength, a quantity that is indicative of the energy that needs to be overcome to propagate earthquake rupture[29–31]. We calculated the breakdown work $W_b$ from the shear strength curve for all the experiments (Fig. 3b and "Methods"). In the fault core materials, the $W_b$ of the experiments performed under fluid-pressurised ($0.06 \pm 0.03$ MJ/m$^2$) and water-dampened ($0.07 \pm 0.01$ MJ/m$^2$) conditions are similar but lower than the $W_b$ ($0.11 \pm 0.01$ MJ/m$^2$) for the room humidity experiments (Supplementary Figure 6a). Clearly, the presence of liquid water activates fluid-driven processes, which reduce the energy required to propagate seismic rupture than thermally driven processes activated under room humidity conditions. The lower $W_b$ in the experiments performed under fluid-pressurised conditions can be explained by the faster initial dynamic weakening and shorter $D_w$ than at other environmental conditions (Supplementary Figure 5a–d). Therefore, fluid-pressurised faults are demonstrably prone to seismic slip if perturbed by the arrival of a seismic rupture. Fluid-pressurised experiments show that fault core materials have the lowest $W_b$ ($0.05 \pm 0.01$ MJ/m$^2$) with respect to hanging wall ($0.08 \pm 0.01$ MJ/m$^2$) and footwall ($0.10 \pm 0.04$ MJ/m$^2$) materials (Supplementary Figure 6b). Despite similar $W_b$ values among Pāpaku thrust materials, the fault core material has the lowest $W_b$ (shortest $D_w$, see Supplementary Figure 5e–h) and is the most favourable to propagate fault slip if perturbed by the arrival of a seismic rupture. The values of breakdown work of Pāpaku thrust materials sheared under fluid-pressurised and water-dampened conditions are comparable with those of clay-bearing materials sampled in the Costa Rica[32] ($W_b = 0.003–0.015$ MJ/m$^2$), Nankai Megasplay[33] ($W_b = 0.05–0.21$ MJ/m$^2$), and the Japan Trench[14,34] ($W_b = 0.003–0.073$ MJ/m$^2$) sheared at similar seismic slip rate, displacement, and total normal stress[35] (Fig. 4c).

In our experiments, because of technical challenges, we applied a low effective normal stress (3 MPa) to the sheared gouges, which are representative of the in situ stress conditions on the Pāpaku thrust. However, effective normal stress is expected to be

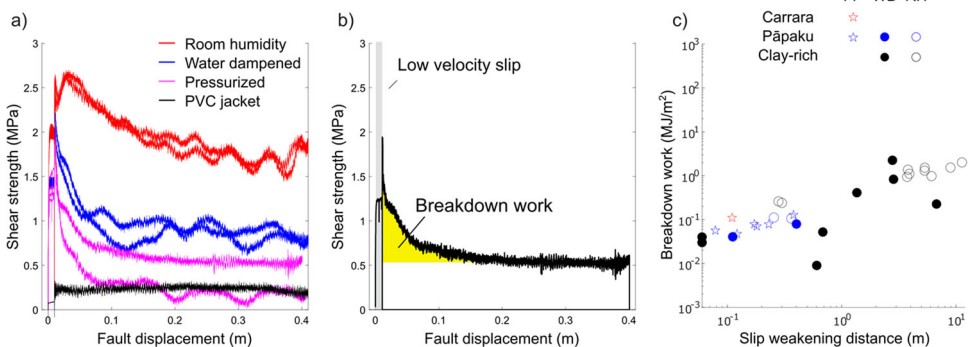

**Fig. 4 Shear strength and breakdown work $W_b$ of the Pāpaku thrust materials. a** The fault core materials of the Pāpaku thrust under fluid-pressurised conditions (magenta lines) have the lowest residual stress and smallest slip weakening distance compared to those deformed under room humidity (red lines) or water-dampened (blue lines) conditions. The strength of the PVC jacket is presented for reference (black line). **b** Breakdown work $W_b$ (yellow shaded area) or energy dissipated to achieve the minimum shear strength during seismic slip controls earthquake rupture propagation: smaller $W_b$, increasing ease of rupture propagation. **c** Breakdown work $W_b$ versus slip weakening distance $D_w$ for Pāpaku thrust materials and Carrara marble gouge with respect to a collection of $W_b$ in clay-rich gouges at seismic slip rates (45–55 wt% phyllosilicates content[35]). Data points are grouped by both lithology and experimental conditions, which are reported as fluid pressurised (FP), water dampened (WD), and room humidity (RH). Our experimental results yield breakdown work comparable with the one measured under similar materials and deformation conditions.

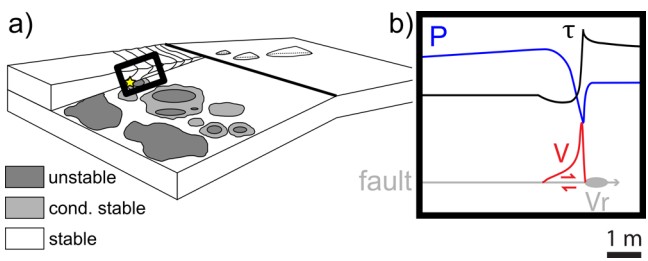

**Fig. 5 Conceptual model of earthquake propagation in shallow subduction faults. a** The subduction interface has a deeper zone with locked and frictionally unstable seismogenic patches (dark grey shaded areas) where tsunamigenic earthquakes (yellow star) can nucleate, surrounded by conditionally stable areas (light grey shaded areas). Around the unstable or conditionally stable areas and especially at shallow depth, the subduction interface is stable (white shaded areas) and undergoes creep (modified from ref. [37]). **b** Conceptual model of seismic slip in the conditionally stable regime. Behind the process zone of the seismic rupture (grey ellipse) propagating towards the right (into neighbouring rocks) at velocity $V_r$ (~1 km/s), the fault slip velocity increases abruptly to ca. 1 m/s (red curve), while the fault strength decreases (dynamic fault weakening, black curve). At the same time, pore fluid pressure inside the slip zone (blue curve) decreases by shear-induced dilatancy and then increases owing to pressurisation processes. The experimental evidences reported in this study suggest that the lower permeability of the clay-rich fault gouges typical of the shallow sections of subduction zones, promotes more rapid pore fluid pressure increase, faster dynamic fault weakening, and lower breakdown work promoting rupture propagation and tsunamigenic earthquakes.

low in shallow sections of clay-rich accretionary wedges of subduction zones[36] and the experimental conditions discussed in this study are likely representative of this natural setting.

Current models of upper subduction zone interfaces include frictionally unstable (where earthquakes nucleate), conditionally stable (associated to SSEs), and stable (associated to fault creep) domains[37] (Fig. 5a). Seismological and geophysical observations[38], in addition to numerical models[15,39], suggest that stable domains with low coseismic shear strength can facilitate seismic rupture propagation. This is also the case for clay-rich sediments, which are usually found in these stable areas[13,14,32,40]. Indeed, at the Hikurangi subduction zone, the IODP Expedition 375 drilled the shallowest part of the Pāpaku thrust branching off the plate decollement above the zone where SSEs are recorded[5] and near the rupture area of the two 1947 tsunami earthquakes[3]. Here, the upward propagation of ruptures during tsunami or megathrust earthquakes through the stable clay-rich domain and towards the seafloor is favoured by the low $W_b$, especially for fluid-pressurised gouges, where low coseismic fault strength and very short $D_w$ are observed (Fig. 5). Fault core and wall rocks display similar $W_b$, possibly indicating that seismic slip at shallow depth can propagate in any of these materials. Moreover, it is possible that, due to the low permeability of shallow sediments in the surrounding of the Pāpaku splay thrust, the pore fluid pressure wave developed during seismic slip remains trapped in the fault core. Depending on the local hydrologic conditions, it may result in a prolonged weakening of the fault and might affect the magnitude of initial afterslip and the occurrence of post-seismic creep or SSEs after the earthquake mainshock.

In this study, by exploiting a novel experimental set-up, we measured the pore fluid pressure variations during simulated seismic slip in non-lithified fault gouges, independently of the dynamic fault strength. We show, for the first time, the coseismic interplay between shear-induced dilatancy and fluid pressurisation in controlling fault strength on sheared materials from a natural subduction zone. We conclude that, during seismic slip and depending on the loading conditions, the initial shear-induced dilatancy can be overcome by a combination of compaction-driven mechanical and thermal fluid pressurisation. Future experiments under constant volume conditions will assess how changes in the initial state of stress, pore pressure, gouge thickness, composition, and permeability among the subducted sediments will affect the coseismic evolution of dilatancy, fault strength, and pore fluid pressure leading to rupture propagation or arrest. Nevertheless, once initial shear-induced dilatancy is overcome, fluid pressurisation in impermeable clay-rich sediments is likely to promote the propagation of seismic ruptures at shallow crustal depths in clay-dominated subduction zones.

## Methods

**Starting materials**. We selected samples obtained during IODP Expedition 375 from Site U1518 belonging to the units 12R1 (hanging wall), 14R1A (fault core), and 21R3 (footwall). The rock samples were dried at 50 °C overnight, then disaggregated with a pestle and mortar, and sieved. The grain size fraction <250 μm was used for the permeability and friction experiments.

The mineralogical composition of these materials in normalised abundances is of 41.8–46 wt% total clay minerals, 27.4–29.6 wt% quartz, 16.2–17.6 wt% feldspars and 9.7–11.4 wt% calcite[18].

**Permeability measurements with the pore fluid oscillation method**. Permeability was measured in a silicon oil-confined permeameter[41] (Supplementary Figure 1) using the pore fluid oscillation method[42–45]. Samples were prepared for testing by sandwiching the gouge material inside a Viton jacket between two porous steel plates. The sample was inserted into the sample assembly, with up- and downstream pore fluid pressure access, and was subsequently placed into a pressure vessel. Because of the sample assemblage, the flow direction was perpendicular to the gouge layer base.

Here, we pressurised the oil using an air-driven pump to apply a confining pressure ($P_c$) around the sample. In the beginning, saturation was performed at low pressure ($P_c = 2$ MPa) to facilitate the equilibration of pore fluid pressure ($P_f = 1$ MPa) across the sample. Following initial saturation, the confining pressure and pore fluid pressure were simultaneously increased to 5 and 3 MPa, respectively, maintaining a maximum pressure differential of 2 MPa to avoid over consolidation. Pore fluid pressure was kept constant (=3 MPa) in the subsequent steps of the experiment, whereas the confining pressure was increased to measure permeability at effective stresses ($P_c - P_f$) of 2, 3, 10, 20, 30, 40, 60, and 100 MPa (Supplementary Table 1).

At each step in confining pressure, equilibration of pore fluid pressure was performed allowing drainage from the upstream side only, waiting for a constant volume in the upstream reservoir. Then, we applied an oscillating upstream pore fluid pressure with a specific amplitude and period (Supplementary Table 1). As the downstream side was under undrained conditions, the downstream pore fluid pressure oscillated with an identical period, but a smaller amplitude and a phase shift with respect to the upstream.

To retrieve the phase shift ($\theta$) and amplitude ratio ($A$) between the up- and downstream pore fluid pressure oscillations, we utilised a fast Fourier transform algorithm. The parameters $A$ and $\theta$ were related to dimensionless permeability ($\eta$), and dimensionless storativity ($\xi$) with the equation[42]:

$$Ae^{-i\theta} = \left( \frac{1+i}{\sqrt{\xi\eta}} \sinh\left[(1+i)\sqrt{\frac{\xi}{\eta}}\right] + \cosh\left[(1+i)\sqrt{\frac{\xi}{\eta}}\right] \right)^{-1} \quad (1)$$

This equation was solved to satisfy the values of $A$ and $\theta$. Initial guesses of $\eta$ and $\xi$ were taken from a lookup table, and a solution was obtained using the gradient descent method to a first-order tolerance of $10^{-10}$.

From dimensionless permeability $\eta$ and dimensionless storativity $\xi$, we calculated permeability ($k$) and pore compressibility ($\beta$) respectively, according to (Eqs. 2 and 3):

$$k = \frac{\eta \pi L \mu_f \beta_D}{ST} \quad (2)$$

$$\beta = \frac{\xi \beta_D}{SL} \quad (3)$$

where $S$ was the sample cross-sectional area (0.0597 m$^2$), $T$ the period of the sinewave (s), $L$ the sample thickness, $\mu_f$ the viscosity of pore fluid ($9 \times 10^{-4}$ Pa·s at 3 MPa and 298 K[46]), and $\beta_D$ the downstream reservoir storage capacity (m$^3$/Pa).

The downstream reservoir storage was calculated as:

$$\beta_d = V_d / P_{p,u} \quad (4)$$

where $V_d$ is the downstream volume (1900 mm$^3$) and $P_{p,u}$ the upstream pore fluid pressure.

**High-velocity friction experiments: the new confined and pressurised fault gouge set-up**. The high velocity friction experiments were performed in a Slow to High Velocity Apparatus (SHIVA)[47] (Supplementary Figure 3a). SHIVA comprises a rotational shaft connected to two electric engines and a static shaft connected to an axial load piston. Samples were deformed in a central sample chamber. On the axial side, the normal force was imposed by an electromechanical piston, measured with a load cell in line with the sample axis, and servo controlled. Torque on the static column was measured with an S-beam load cell attached to an arm. Normal stress was calculated by dividing the normal force by the nominal sample contact area and torque was converted to shear strength[48]. The displacement normal to the gouge layer (i.e. axial displacement) was measured with two LVDTs. A high range low-resolution LVDT (50 μm resolution and 50 mm stroke) was placed on the static side of the machine next to the normal force load cell. A low range high-resolution LVDT (0.03 μm resolution and 3 mm stroke) measured displacement between the stationary sample holder and the sample chamber wall. The high-resolution measurement of axial displacement was used to monitor the gouge layer thickness change during the experiments. The rotation of the rotary shaft was imposed with two electric motors: a small 5 kW motor and a larger 280 kW motor. To impose slip velocities from $\sim 7 \times 10^{-6}$ to 0.0026 m/s (i.e. 0.004 to 1.5 rpm converted into our sample area of 0.00205 m$^2$) we used the smaller motor, and for slip velocities from 0.0026 to 5.35 m/s (1.5 to 3000 rpm converted into our sample area of 0.00205 m$^2$) we used the larger motor. Angular rotation was measured with two optical encoders, one with finer (629760 div, ~0.17 μm) and one with coarser

(400 div, ~267.5 μm) resolution. Above 0.15 m/s the fine encoder saturated and all reported measurements above this velocity were derived from the coarse encoder. By combining the two measurements we obtained the incremental displacement and by extension velocity, using numerical derivation with respect to time.

The sample holders comprised two cylinders of radius $r = 0.02555$ m (contact area of 0.00205 m$^2$). Porous steel plates (Mottcorp, grade 0.5, permeability of $\sim 1 \times 10^{-12}$ m$^2$) were inserted into tolerance fit sockets, designed with small channels connecting to the pore fluid pressure line to improve fluid distribution (Supplementary Figures 3b and 4a). Before each experiment, the static sample holder was jacketed with a polyvinyl chloride (PVC) shrink tube fixed to the static holder with stainless-steel wiring. The fault gouge was weighed (8.00 g) and evenly distributed on top of the entire stationary sample assembly. At this stage, the sample thickness was measured with a calliper with values between 4.5 and 4.75 mm. Deionized water was added on top of the gouge layer to dampen it and allow installation of the assembly in the horizontal sample chamber. The assembly was inserted into a pressure vessel and subsequently loaded into SHIVA. Following this, the axial piston was used to insert the rotary sample holder into the pressure vessel. The internal surface of the jacket was lubricated with MoS$_2$-based solid lubricant to reduce friction between the rotary sample holder and the PVC jacket. A water-filled pressure vessel[49] was used to provide confining pressure ($P_c$) controlled with a syringe pump. Confining pressure impeded the extrusion of the gouge layer during seismic slip (Supplementary Figure 3b). Pore fluid pressure ($P_f$) was introduced into the gouge layer from the stationary side and was controlled using an independent syringe pump (Supplementary Figure 3b). Pore fluid pressure was monitored on the rotational (downstream) side with a pressure transducer and on the static (upstream) side with a miniaturised piezoresistive transducer (Keller, 2MI-PA210) under the porous plate (~3.85 mm from the edge of the gouge layer, Supplementary Figure 4a). The downstream reservoir had a volume of 4.67 mL; therefore, the ratio of the pore to downstream reservoir volume was in the range 0.28–0.78. To monitor temperature during the experiments, a K-type thermocouple was placed inside the static sample holder with the tip at ~1 mm from the static boundary of the gouge layer (Supplementary Figure 4a).

Fluid-pressurised experiments were performed on Pāpaku thrust materials. Each experiment started with a gradual and simultaneous increase of the normal stress ($S_n$) and confining pressure ($P_c$) followed by an increase of fluid pressure ($P_f$). Due to the low permeability of the Pāpaku thrust materials, we achieved saturation at $P_c = S_n = 1.25$ MPa and $P_f = 0.25$ MPa. To ensure saturation, we waited until the downstream pressure was equilibrated with the upstream pressure and the pump volume remained relatively constant. The air within the pressure line and the sample was evacuated by pumping pore fluid with the downstream valve opened. After evacuation and saturation, we applied the final values of $S_n = 6$ MPa, $P_c = 4$ MPa and $P_f = 3$ MPa, once again waiting for equilibration of pore fluid pressure and pump volume. Then, the samples were sheared at $10^{-5}$ m/s for 0.01 m to compact the gouge. After this stage, the shear stress acting on the fault was released. Then, after pore fluid pressure equilibrated, we imposed one seismic slip pulse with an acceleration and deceleration of 5.7 m/s$^2$, velocity of 0.8 m/s and displacement of 0.4 m. The experiments were repeated twice for each type of fault gouge material to obtain data reproducibility. To test if the measured pore fluid pressure variations were independent of the experimental configuration and sample assemblage, we performed a series of experiments with the more permeable Carrara marble gouge (~99 wt% calcite and 1 wt% quartz, grain size <250 μm). The Carrara gouge was tested under fluid-pressurised conditions, following the same experimental operations and boundary conditions applied to the Pāpaku thrust materials (Fig. 3). During the seismic velocity pulse, pore fluid pressure was controlled at a constant value on the upstream side, controlled at a rate of 1 Hz. However, since the seismic slip pulse duration was <1 s, the response of the control system enhanced the sample-related dilation at the onset of the seismic slip pulse.

Additional experiments were performed on the Pāpaku thrust fault core materials under room humidity and water-dampened conditions. To impose room humidity conditions, fault gouge was tested without adding water. To impose water-dampened conditions, 4 mL of deionized water (50 wt%) was added to the gouge layer so that it was only partially saturated. After installation into the sample chamber, we imposed to the gouge layer a normal stress of 3 MPa and a confining pressure of 1 MPa, whereas pore fluid pressure was neither imposed nor controlled and the fluid lines were kept open to the atmosphere (0.1 MPa). Then, the samples were sheared at $10^{-5}$ m/s for 0.01 m to achieve compaction and then we imposed one seismic slip pulse with an acceleration and deceleration of 5.7 m/s$^2$, velocity of 0.8 m/s and displacement of 0.4 m. The control experiments were repeated twice for each condition to obtain data reproducibility.

A control experiment was performed on the lubricated PVC jacket used to isolate the gouge layer from the confining medium without any gouge layer. The jacket was tested at $S_n = P_c = P_f = 0$ MPa by imposing, in sequence the low-velocity slip at $10^{-5}$ m/s for 0.01 m and the seismic slip pulse with an acceleration and deceleration of 5.7 m/s$^2$, velocity of 0.8 m/s and displacement of 0.4 m. During all velocity pulses, the separation between the two sample holders was set to ~0.004 m, equivalent to the maximum thickness of the gouge layer that was tested in our pressurised experimental set-up.

**Estimation of the hydraulic diffusion time**. We calculated the hydraulic diffusion time across the gouge layer during the seismic slip velocity experiments as $t_{hy} = d^2/$

$\alpha_{hy}$, where $d$ is the gouge layer half-thickness (m) and $\alpha_{hy}$ is the hydraulic diffusivity (m$^2$/s). Hydraulic diffusivity is defined as $\alpha_{hy} = k/(\mu_f \, \varphi \beta_f)$, with $k$ the permeability of the gouge layer at 3 MPa effective stress (~$4.28 \times 10^{-20}$ m$^2$ for Pāpaku thrust materials and ~$3.25 \times 10^{-17}$ m$^2$ for Carrara marble gouge[26,50]), $\mu_f$ water viscosity ($9 \times 10^{-4}$ Pa·s), $\beta_f$ water compressibility ($4.44 \times 10^{-10}$ 1/Pa) and $\varphi$ the porosity of the gouge layer. We considered the gouge layer half-thickness $d$ in the range from 0.0012 (the smallest half-thickness after seismic slip) to 0.0017 m (the largest half-thickness before seismic slip). We considered a minimum porosity of 20% and a maximum porosity of 40%. The hydraulic diffusion time varied between 26.9 and 107.8 s for Pāpaku thrust and between 0.03 and 0.14 s for Carrara marble gouges.

**Estimation of the temperature rise in the gouge layer**. We calculated the maximum temperature rise at the centre and edge of the gouge layer and compared the latter to the temperature measurements during the high-velocity friction experiments (Supplementary Figures 7 and 8). To estimate temperature, we solved the diffusion equation:

$$\frac{\partial T}{\partial t} = \frac{\kappa}{\rho c}\frac{\partial^2 T}{\partial x^2} + \frac{\tau(t)\dot\gamma(x,t)}{\rho c} \tag{5}$$

where $x$ (m) was the direction perpendicular to the gouge layer, $\kappa$ thermal conductivity (W/(m·K)), $\rho$ density (kg/m$^3$), $c$ heat capacity J/(kg·K) and $\tau(t)$ the experimental measurement of shear strength (Pa). The gaussian function $\dot\gamma(x,t)$ described the shear strain rate as[10]:

$$\dot\gamma(x,t) = \sqrt{\frac{2}{\pi}}\frac{V(t)}{w}\exp\left(-\frac{x^2}{w^2}\right) \tag{6}$$

where $V(t)$ was the measured slip velocity during the experiments (m/s) and $w$ the thickness of the localised zone (set at $10^{-4}$ m).

The partial differential equation in (Eq. 5) was solved using a 1D time-dependent finite difference numerical model using the implicit scheme formulation. The $x$ domain was defined by 501 nodes, with constant grid spacing, and organised into three subdomains with variable conductivity, density and heat capacity. The gouge subdomain was set in the interval $-d \leq x \leq d$, with $d$ the gouge layer half-thickness measured before imposing the seismic slip velocity pulse (i.e. $d$ ranged from 0.0012 to 0.0017 m). The gouge subdomain had a thermal conductivity of 1.5 W/(m·K), a density of 2400 kg/m$^3$ and a heat capacity of 1000 J/(kg·K)[51]. Two subdomains, representing the steel porous plates, were set in the intervals $-d - 0.02 \leq x \leq -d$ and $d \leq x \leq d + 0.02$, with a thermal conductivity of 8.6 W/(m·K), density of 6162 kg/m$^3$ and heat capacity of 475 J/(kg·K) (Mottcorp datasheet).

The initial temperature was set to 25 °C in all model nodes. The boundary conditions were imposed setting a constant temperature $T = 25$ °C in the first (upstream) and last (downstream) nodes of domain $x$. The measured temperature during the high-velocity friction experiments was replicated by the estimated temperature in the corresponding position of the thermocouple (i.e. ~1 mm away from the gouge layer, in $x = -d - 0.001$, see Supplementary Figures 7 and 8). The maximum estimated temperature in the centre of the gouge layer (i.e. $x = 0$) during the seismic velocity slip pulse was used to discuss the active pressurisation mechanisms governing dynamic weakening (Supplementary Table 2).

**Estimation of the fault strength constitutive parameters and the breakdown work**. The analysis of the dissipated energy (or breakdown work $W_b$) during earthquake propagation was performed using the measurements of shear strength with displacement during the seismic velocity slip pulses[29,32] following the definition of $W_b$ from seismological studies[30,31]. In this framework, the experimental measurements of fault strength represent the maximum shear stress that the fault can sustain.

First, we described the measured evolution of shear strength $\tau$ with displacement $s$ using a best-fit procedure with the following power-law function:

$$\tau(s) = \tau_r + \left(\tau_p - \tau_r\right)\exp\left[\log(0.05)\left(\frac{s}{D_w}\right)^{\alpha}\right] \tag{7}$$

where the best-fit parameters were defined as the peak strength $\tau_p$, residual shear strength $\tau_r$, slip weakening distance $D_w$ and exponent $\alpha$ (Supplementary Table 2).

The experimental measurements of shear strength were filtered by applying a moving average filter (having a span of 1000 points). Then, the breakdown work $W_b(s)$ was obtained as function of displacement $s$ as:

$$W_b(s) = \int_0^s (\tau(s) - \tau_{min}(s))\mathrm{d}s \tag{8}$$

with $\tau_{min}(s)$ defined as is the minimum strength in the displacement interval from 0 to $s$. The breakdown work $W_b$ discussed in this paper, is defined for a final displacement $s = 0.4$ m in each seismic slip velocity step.

## Data availability

Correspondence and request for additional materials should be addressed to stefano.aretusini@ingv.it. All the experimental raw data are available in Zenodo with the identifier "https://doi.org/10.5281/zenodo.4268280".

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

## Acknowledgements

We acknowledge the ERC Consolidator Grant 614705 NOFEAR. S.A. acknowledges C. Cornelio for insightful discussion, B. Carpenter and M. Scuderi for their suggestions on improving the fluid-pressurised experimental set-up, and G. Romeo for developing the miniaturised pressure acquisition system.

## Author contributions

All the authors discussed the concepts proposed in this study. S.A., E.S. and G.D.T. designed the new pressurised-fluid sample holder. S.A., E.S. and C.W.H. performed the high-velocity friction experiments. S.A. ran the numerical models. C.W.H. performed the permeability measurements and relative data reduction. F.M. provided the experimental samples, the geological and mineralogical data. S.A. wrote the manuscript and all the authors contributed to the final revisions.

## Competing interests

The authors declare no competing interests.
