## [Peer Review File · Nature Communications]

REVIEWER COMMENTS

Reviewer #1 (Remarks to the Author):

Review of the paper, "Fluid pressurisation and earthquake propagation in the Hikurangi subduction zone", by Aretusini and others.

This manuscript reports experimental results on fluid pressurization during coseismic slip under partially drained condition. The authors used impermeable fault materials drilled from Hikurangi subduction zone and found that these samples showed a drastic weakening due to thermal pressurization during a coseismic slip. The manuscript is well written, however, it's difficult for me to evaluate whether this manuscript is worth to be published in Nature communication because this manuscript is not more than a case study for Hikurangi subduction zone to compare other previous studies (Ujii et al., 2013; Rempe et al., 2017; Badt et al., 2020, etc.) and there is a technical issue. I understand huge efforts by the authors for this study, but the authors faced typical (mostly technical) issues on these kinds of high velocity friction experiments and failed to overcome them. Conclusion is not surprising regarding the previous study for Tohoku subduction zone (e.g., Ujii et al., 2013). So the result and the conclusion in this study is applicable for the most of subduction zones, but it have been already reported in previous studies. I would recommend the authors to submit this manuscript to a more specific journal. I don't point one by one for specific comments at this stage, but I would be happy to make further comments if the editor allows the author to revise the manuscript.

1. The experimental design is needed to be improved. The experimental condition in this study is not under ideal drained condition or undrained condition. The pore pressure is controlled but only partially. Even the upstream pore pressure is not satisfactorily controlled. I think the delay of the pore pressure (Fig. 2) is not due to the low permeability of the sample but just clogging of the porous plate placed both side of the sample (meaning artificial problem). This makes the interpretations of the experiments more complex. New pore pressure system developed by authors should be appreciated, but more standard sample assembly (drained/undrained conditions using permeable/impermeable sample block; e.g., Ujii et al., 2014) could be better in terms of evaluating the effect of the pore fluid pressurization.
2. I'm curious how the friction coefficient (shear stress divided by effective normal stress) evolves during the slip. Figure 2 shows that P_f changes >0.5 MPa during the slip, while the shear stress decays relatively smooth. This means that, if measured P_f s are correct, the friction coefficient increased during the slip while the value of the friction coefficient is still small.
3. As the authors pointed in the manuscript (lines 131–137), there are several mechanisms to increase the pore pressure during the coseismic slip. The authors have enough dataset to distinguish which mechanism can be the dominant for the weakening (shear stress, permeability, porosity change, temperature change, storage capacity, etc). It's not shown in the manuscript but I guess the axial shortening (sample thickness during the experiment, or at least before/ after experiment) is also measured (LVDT is there in Fig. S3a), so they can evaluate the amount of the compaction during the coseismic slip. In my opinion, this kind of analyses is highly required for publishing the research to high impact journals.
4. The pore pressure drop at the initial 0.1 s can be by the fracturing and dilation of the gouge layer. This makes sense to me because fracture energy can be calculated from the decay of the shear stress and the ISCO pump doesn't have capability to keep the pressure constant at such fast change of the pressure.
5. Time delay of the pore pressure recovery can be due to the clogging of the porous plates, may not be due to the low permeability of the gouge. I guess "porous" in the manuscript means just there are several holes on the plates, not porous material like a sandstone. As the authors described in lines 318–326, 100s after shearing is long enough for re-equilibration of the pore fluid pressure between

upstream and downstream lines. Using the same equation, if you calculate the diffusion length of fluid L_c , L_c will be 1 cm after 100 s, which value is more than twice of the sample thickness. So delay of the pore pressure recovery is probably the technical issue, not from the sample. Technically the authors can evaluate the effect of the clogging of the porous plates on the pore pressure change if they run an additional experiment using clayey gouge then measure permeability of the clogged porous plates.

Other comments

I don't think the ISCO syringe pump has enough response for such high speed experiments since the pump can be programmable only up to 1 Hz or even lower. The authors might replace this pump to an air reservoir and air/fluid separator pressurization system for this range of the pressure.

Lines 135–136: thermal dehydration or something equivalent would be better, otherwise add Han et al. (2007) for the case of thermal decarbonation of marble.

Some data of pore compressibility is not well solved (e.g., 3rd and 6th row in Table S1). These data must be from the failure of the numerical fitting and not be the real sample data.

Reviewer #2 (Remarks to the Author):

This is a nice study in which high velocity frictional behaviour of fault gouge recovered from the Hikarangi Trough is tested under fluid saturated, confined conditions. The novelty of the study is not so much in the findings of the study that mirror previous work done, but for the fact that this is the first time that these materials have been confined and tested at seismic slip velocity. Consequently the study verifies and confirms the conclusions drawn in previous studies. I think the authors acknowledge this well and it is good that the results are presented in a balanced way and not oversold. The study will be of wide interest to a range of geoscientists interested in subduction zone seismicity.

I think that the manuscript could do with some polishing. I would suggest that the authors think about fluid pressure diffusion timescales (characteristic times) that would probably show that characteristic times are much longer the duration of the experiments and therefore it is not that surprising that the experiments that they ran give very similar results to the unconfined equivalents ran previously as the mechanical data will be most influenced by the pore pressure built up within the layer which does not have enough time to escape from the layer whether or not the experiment is confined. I think this would be a useful addition and it wouldn't detract from the fact that these are still very technical experiments that have been performed and they do still provide plenty of useful insight into the physics of slip.

I have some specific comments that are detailed below that I hope might improve the clarity of the manuscript in places.

I am happy to clarify with the authors anything that is not clear in my comments.

Dan Faulkner, University of Liverpool, UK.

Line 36. I feel that this may be posed a little more clearly. An important point for me, summarized in the 2011 paper, is that clay-rich forearc sediments should be (are) velocity strengthening and consequently will kill off any ruptures that propagate into this region (this is illustrated by the

seismically quiet zone typically seen from the surface to ~10km depth in subduction forearcs).

The authors say that co-seismic weakening will reduce the mechanical work, which is true, but the key factor is the co-seismic weakening results in overall velocity weakening behaviour which is the prerequisite to the rupture propagating at all. The comment applies to line 47, where I feel it should state that the co-seismic weakening allows for the possibility of rupture propagation and the small energy barrier (presumably from the low breakdown work/fracture energy) will promote large slip.

Line 65. It would be good for clarity to define what is meant by 'ductile' and 'brittle' here. These terms are often used loosely and mean different things to different people.

Line 80. This sentence needs modification. While experiments may be a guide to the shear stress evolution during an earthquake, the actual evolution is a highly coupled process that spontaneously evolves according to nature's boundary conditions, which are certainly not the same in a controlled velocity experiment. Hence the experiments will be a guide to the shear stress evolution, but not equivalent.

Line 90. How fast could the upstream pump react? Can it maintain the upstream pore fluid pressure over fractions of a second?

Line 91. What was the volume of the downstream reservoir? This is key to report as any pressure change here will depend on the size of the reservoir. In fact, the ratio of the pore volume of the sample to the downstream reservoir volume will be very useful to estimate.

Figure 2. These are great results. I am particularly intrigued by the drop in pore fluid pressure as the velocity increases. This is presumably shear enhanced dilatancy (although this depends on how fast acting the pump is, see comment for line 90) which (I think) would be the first time that this has been observed in a HV friction experiment.

The immediate recovery of this pressure change is also interesting. The downstream recovers much faster. Why do you think that is? Is it because the slip has localized in the interface with the downstream reservoir? Or is it a pump-related effect?

Line 104. The longer-term recovery would obviously be much slower in the downstream (uncontrolled) than the upstream. If the response of the upstream pump is immediate (see comment for line 90) then the upstream pressure would sit perfectly at 3MPa.

Line 122. I think this comment requires a little more detail before it can be useful to the reader. Do you mean that the experiments on Carrara marble was pretty much drained (i.e. very high permeability), and this is how the pore fluid pressure variations were correlated to the shear stress? If so then it may be worth referencing Faulkner et al. 2018 JGR here, where there are relations on the characteristic timescale for fluid diffusion, given the hydraulic diffusivity. A comparison between the characteristic timescales for the Carrara experiments and the gouge experiments may be useful here (using l^2/κ).

Line 129. I think the terminology could be tighter here. Note that it is the pore fluid pressure that diffuses – not strictly the water. Also diffusion occurs from high pressure to low pressure, so the fluid pressure will be diffusing from the within the layer out into the pore reservoir volumes.

Line 136. This reference needs formatting in the journal style.

Line 144. How can vaporization contribute toward pore fluid pressurization? As the temperature increases, the water may start to undergo a phase transformation – this will buffer the temperature – but then won't the water vapour be at the same pressure as the liquid water before the phase transformation? If the water pressure did start to increase would this not buffer the transformation so that more temperature would be required before further transformation can occur?

Line 144. It is likely that only the smectite will breakdown at these temperatures, but I guess that's enough.

Line 158. I have the same comment as I had for line 80 – the W_b is not a material property and will depend on the boundary conditions of the experiment and hence it is only a guide to what happens at a propagating rupture tip.

The discussion here could also be usefully compared to a recent compilation of w_b values by Seyler et al. 2020 EPSL.

Line 173. This is a problem that all of us have to face – the low normal stress in the experiment is representative of a couple of hundred metres at most – unless overpressured fluids are present.

Reviewer #3 (Remarks to the Author):

This article reports on laboratory high-velocity friction experiments focusing on samples from a splay fault in the Hikurangi subduction zone. In previous high-velocity friction studies, the lack of direct pore pressure measurement was a major drawback, which has been solved here and is an important step which deserves publication.

I do not see anything "wrong" with this study; the data look good and the interpretations are reasonable. I am worried that this study is actually too comprehensive for a short-format journal. Since these are the first measurements using the new technique, they need to be carefully documented. A lot of work has been done for this and this results in several different "sub-studies": the comparison between room-dry, wet, and pressurized experiments, the permeability measurements, the comparison between Carrara marble and the Hikurangi samples, and applying the Hikurangi data to the Hikurangi margin itself. I think that both the methods and the data (all the data) deserve to be extensively presented and discussed in a real article and not be tucked away as supplementary material. I certainly understand the desire, or the pressure, to publish in high impact short-format journals, but think that by cramming too much material into a short article that both the data presentation and the discussion suffer.

Other comments:

In Lines, 112-115, a point of emphasis seems to be that the fault core contrasts with the wall rock, but looking at the data in the supplementary material this is a bit misleading, because the behavior of the fault core does not seem very different compared to the wall rocks. For example, it is mentioned that the slip weakening distance is smallest for the fault core, but it is close to the hanging wall values. The residual friction of the fault core is actually higher than that of the wall rocks. The breakdown work is also not very different. I think it is worth acknowledging that properties of the fault core are not much different from the wall rocks, which is consistent with structural observations from the cores, and has some important implications for the distribution of slip at shallow depths.

I think it is actually important to include the Carrara marble data in the main text. This is because the

technique is new, so data needs to be presented on a well-studied standard, and not just on a specific material from a particular field site, which may be unique or unusual.

The article focuses on the pore pressure rise, but I think the pore pressure decrease at the beginning of the experiment is just as important, and something that is rarely discussed. This decrease clearly indicates dilatancy, and looking at Figure 2, it can be seen that by the time the average pore pressure has returned to the original 3 MPa most of the dynamic weakening is already completed. This is important because it seems to argue against thermal pressurization as a cause of dynamic weakening. Also, the dilatancy looks like it drops the pore pressure from 3 to about 2.2 MPa, which is a significant increase in effective stress. Since dynamic weakening occurs anyway, this maybe suggests that dilatant hardening – suggested as a mechanism for SSEs, and of course Hikurangi is famous for SSEs – is not effective. I know that it is popular to discuss pore pressure increases, but in my view the pore pressure decreases are just as important and both need to be discussed.

Moving on to the pore pressure increase, it seems like an important factor here is that the pore pressure increase is long-lasting, much longer lasting than the slip pulse here, and by extension probably much longer than a (main) fast slip event in the Hikurangi. It seems to be important because it implies that the fault is weakened for an extended period of time following a slip event, but it is not discussed.

Which brings me to my final main comment, that these experimental results aren't very well connected to the Hikurangi subduction zone and to the specific behaviors that occur there. As a side note, I have to point out that it is unknown if SSEs or tsunami earthquakes actually occurred on the fault drilled here. But anyway, the conclusion here is that thermal pressurization can contribute to the propagation of earthquakes or tsunami earthquakes to shallow depths. Ok, in general that is true. But does this apply to SSEs? Ordinary earthquakes? Tsunami earthquakes? All three? And it is stated that W_b and D_w are small. This needs to be qualified somehow - compared to what? Marble? Or other subduction zones? A good approach could be to compare these measured values with the comparison published recently by Seyler et al. (2020 EPSL).

Line 10: I suggest shortening the text here and not referring to the Tohoku earthquake, because the reader might mistakenly think the paper is about Tohoku. It's fine to have this in the introduction, just not in the abstract.

Line 41: "non-lithified" is better than "non-cohesive" here

Line 60-61: I don't think it is known that the Papaku fault has hosted any kind of tsunami earthquake or SSE, especially at the drill site. Or do you mean the plate boundary fault? Make this clearer.

Line 73: are the permeability measurements done on intact samples? How were they oriented relative to the flow direction?

Line 119-120: what was the maximum decrease in pore pressure?

Line 194: Figure 4b is confusing to me. First, it would be helpful to distinguish better the rupture velocity and the slip velocity. Second, behind the rupture front is denoted as "slipping" but at what velocity? if V has dropped to zero, it shouldn't be slipping. Is the shear stress then low because it is at its dynamic value? Or is it because the fault is unloaded? To the far left, the pore pressure pulse has worn off, but if the fault is slipping shouldn't the pressure still be elevated?

REVIEWERS COMMENTS

Reviewer #1 (Remarks to the Author):

Review of the paper, “Fluid pressurisation and earthquake propagation in the Hikurangi subduction zone”, by Aretusini and others.

This manuscript reports experimental results on fluid pressurization during coseismic slip under partially drained condition. The authors used impermeable fault materials drilled from Hikurangi subduction zone and found that these samples showed a drastic weakening due to thermal pressurization during a coseismic slip. The manuscript is well written, however, it's difficult for me to evaluate whether this manuscript is worth to be published in Nature communication because this manuscript is not more than a case study for Hikurangi subduction zone to compare other previous studies (Ujiie et al., 2013; Rempe et al., 2017; Badt et al., 2020, etc.) and there is a technical issue. I understand huge efforts by the authors for this study, but the authors faced typical (mostly technical) issues on these kinds of high velocity friction experiments and failed to overcome them. Conclusion is not surprising regarding the previous study for Tohoku subduction zone (e.g., Ujiie et al., 2013).

So the result and the conclusion in this study is applicable for the most of subduction zones, but it have been already reported in previous studies. I would recommend the authors to submit this manuscript to a more specific journal. I don't point one by one for specific comments at this stage, but I would be happy to make further comments if the editor allows the author to revise the manuscript.

We thank Reviewer #1 for his/her remarks. However, we wish to highlight the following points, which are crucial for us and for this study:

1) First, the reviewer is not fully correct when she/he summarizes our study: she/he states “*samples showed a drastic weakening due to thermal pressurization during a coseismic slip*”. Actually, the experimental evidence presented here, where for the first time (as recognized by the other two reviewers) pore fluid pressures and temperatures were measured in fluid-saturated gouges sheared at seismic slip rates, questions the hypothesis that thermal pressurization alone is responsible for fault weakening during seismic slip. Actually, our experimental results indicate that is compaction-driven fluid pressurization and thermal pressurisation which make the fault weak during seismic slip. This finding, noticed by the other reviewers, is relevant for the mechanics of shallow-seated crustal faults.

2) The reviewer states that we “*failed*” to overcome “*a technical issue*” related to “*these kinds of high velocity experiments*”. Failure is a quite strong word, which is at odds with (i) the experimental evidence reported in this study and (ii) the comments of reviewers #2 and #3. In fact, since the Reviewer 1 is not specific here, we assume that she/he is referring to the measurements of pore fluid pressure during seismic slip (e.g., Fig. 2, new Fig. 3). As recognized by the other reviewers, pore fluid pressures were never measured, nor controlled or maintained in any previous experiment performed on fluid pressurized

gouges (including subduction zone non-cohesive materials): this is one of the main motivations of this study and its main added value.

The achievement of the experimental results presented here, obtained thanks to the design of a novel “pressurized” gouge holder took years of work and tests. In fact, both reviewers #2 and #3 acknowledged the novelty of our experimental configuration and results: Reviewer #2 wrote “*this is the first time that these materials have been confined and tested at seismic slip velocity*” and Reviewer #3 stated “*the lack of direct pore pressure measurement was a major drawback, which has been solved here and is an important step which deserves publication*”

3) With respect to the experimental configuration used by Faulkner et al, GRL 2011, Uijie et al., Science 2013, Remitti et al., 2015, the new experimental configuration presented in this manuscript allows measurements of pore fluid pressure independently of the shear stress and normal stress (Fig. 2, new Fig. 3). These independent measurements provide experimental data for future modelling of seismic rupture propagation (e.g., Murphy et al., 2018 EPSL) and to discriminate whether fault weakening results from a reduction of the shear strength or from pressurization and reduction of the effective normal stress. This is a fundamental contribution to understand the deformation processes leading to fault dynamic weakening during earthquakes. As a matter of fact, the same Reviewer #1 acknowledges this as he wrote “*I’m curious how the friction coefficient (shear stress divided by effective normal stress) evolves during the slip*” and, thanks to this dataset, we can discuss this evolution. Lastly, we wish to highlight that the independent measurements of pore fluid pressure and temperature allow us to claim that fluid compaction-driven pressurization together with thermal pressurization have a pivotal role in fault dynamic weakening in the experiments discussed here.

4) This study is much more than a case study specific to the Hikurangi subduction zone. The samples we tested were retrieved directly from the active fault (IODP project 375) and they are representative of a subduction zone active splay fault (Fagereng et al., 2019). These samples were retrieved specifically to allow geophysicists and geologists to understand the mechanics of tsunamigenic earthquakes or silent slip events. Actually, the same Reviewer #1 acknowledged in the same review (see below) that “*the result and the conclusion in this study is applicable for the most of subduction zones*”, thus we are a bit confused by his/her comment.

Therefore, again, we stress the relevance of this study, not only because it reports new findings, but because it also verifies theoretical conclusions poorly supported by previous experimental studies because of the technical limitations discussed at point 2 above and at page 5 of this rebuttal letter.

In the responses below, we also provided a detailed explanation of the technical limitations of our experimental approach, as correctly requested by the Reviewer.

1. The experimental design is needed to be improved. The experimental condition in this study is not under ideal drained condition or undrained condition. The pore pressure is

controlled but only partially. Even the upstream pore pressure is not satisfactorily controlled.

We acknowledge that the ISCO syringe pump cannot keep the upstream pore pressure constant in all the experiments, because of the sample-related effect of dilatancy and rapid evolution of pressure (also acknowledged by the Reviewer, see his/her comment 4 below). However, constant pressure head is ensured and therefore the experimental conditions are drained from the upstream side of the sample before the experiment. The downstream pore fluid pressure is nominally undrained during the high velocity slip due to the low permeability of the gouge material. However, at larger timescales (i.e. after high velocity slip), the sample transitions to a drained condition. We selected this configuration to measure any pressurisation occurring during high velocity slip, which would not be possible under fully drained boundary conditions. Despite this physical limitation, that we acknowledged in lines 332-334:

“Since pore fluid pressure was controlled with a rate of 1 Hz, and the seismic slip pulse duration was < 1s, the response of the control system enhanced the sample-related dilation at the onset of the seismic slip pulse.”

We are still convinced that this was a technical advance. As recognized by Reviewer #2: *“these are still very technical experiments that have been performed and they do still provide plenty of useful insight into the physics of slip”*.

I think the delay of the pore pressure (Fig. 2) is not due to the low permeability of the sample but just clogging of the porous plate placed both side of the sample (meaning artificial problem). This makes the interpretations of the experiments more complex.

We thank the reviewer for allowing us to further clarify this relevant point. The delay of the pore pressure pulse is not due to clogging of the porous plate. In fact, the porous plates are made of sintered stainless steel spheres with controlled porosity and permeability, as we described in the Methods section of the manuscript (L. 297):

“Mottcorp, grade 0.5, permeability of $\sim 1 \cdot 10^{-12} \text{ m}^2$ ”

Porous plates are in common usage in the rock deformation community e.g. Faulkner et al. 2018 JGR or Scuderi and Colletini 2016 EPSL. Before each experiment, the two plates were placed in an ultrasonic bath for two successive runs of 10 minutes to clean them. Instead, the delay of the downstream pore pressure wave is a result of the transport properties of the clay-rich gouge (e.g., permeability $\approx 10^{-20} \text{ m}^2$). This fact is supported by the experimental evidence that, by shearing a much more permeable sample (i.e., Carrara marble gouge, permeability $\approx 10^{-17} \text{ m}^2$), under identical deformation conditions (same initial pore fluid pressure, normal stress, target slip rate, etc.) the pore fluid pressure rise was detected coeval with the seismic slip pulse (see old Suppl. Fig 4, now figure 3 of the main text, and see also new Suppl. Fig. 5c-5d). In conclusion, the measured time delay of the pore pressure pulse is due to the transport and poroelastic properties of the gouge (permeability, compressibility, etc.) and the time delay is not the result of an experimental spurious effect (clogging of the porous plate). In order to highlight this important point, we

moved the Carrara marble experiment to the main text, reinforcing the fact that this experiment was done on well-known more permeable materials to benchmark the capabilities of our novel experimental assembly.

New Figure 3.

New pore pressure system developed by authors should be appreciated, but more standard sample assembly (drained/undrained conditions using permeable/impermeable sample block; e.g., Ujiie et al., 2014) could be better in terms of evaluating the effect of the pore fluid pressurization.

The sample assembly first used by Faulkner et al., 2011 and later by Ujiie et al 2013 consisted of more permeable (Berea sandstone) or less permeable (gabbro) rock cylinders sandwiching the gouge layer. A few mL of water was used to saturate the gouge and the gouge and the rock cylinders were then confined with Teflon ring. The sample assembly used in these previous studies did not allow the authors to measure the pore fluid pressure. This crucial limitation motivated our research and led to the design of the new sample assembly presented here. We chose to use permeable porous plates to facilitate the measurement of the pore pressure rise in the gouge layer (see also the previous comment). In the case of gouge deriving from the Hikurangi materials, the permeability is 10^{-20} m^2 with diffusion times of $\sim 30 - 100 \text{ s}$, (see the Method section of the main text). This diffusion timescale is 2-3 orders of magnitude larger than the duration of the slip pulse ($\sim 0.5 \text{ s}$). This temporal delay prevents a concurrent measurement of the pore pressure inside the gouge layer, independently of the forcing block permeability.

In conclusion, we do not agree that the setup used in the past is better than the one proposed here. We would write that is different, not better. In fact, the previous gouge holder did not allow the experimentalist to control or even measure the pore fluid pressure either far or close to the experimental slipping zone. Moreover, the sample assemblage used in the past, as well discussed in Sawai et al. (2012) had several issues including: i) possible uncontrolled escape of pressurized water from the sample, ii) spurious friction on

the lateral surface of the rock forcing cylinder resulting from the Teflon confinement (especially if rock debris end in the interface between the Teflon ring and the solid rock), iii) Teflon breakdown at temperatures above 180 °C which results in fluorine release that dissolves the minerals in the gouge (so the microstructure cannot be compared to natural ones where fluorine release is absent, see De Paola et al., 2011 for discussion), iv) the previous sample assemblage only allowed measurements of the apparent friction coefficient i.e. did not allow to constrain the value of the friction coefficient due to the lack of independent pore fluid pressure measurements.

2. I'm curious how the friction coefficient (shear stress divided by effective normal stress) evolves during the slip. Figure 2 shows that Pf changes >0.5 MPa during the slip, while the shear stress decays relatively smooth. This means that, if measured Pfs are correct, the friction coefficient increased during the slip while the value of the friction coefficient is still small.

The Pf curve reported in figure 2 of the manuscript is the pore pressure measured at the downstream boundary of the sample assembly. This Pf measurement yields the closest estimate of the pore fluid pressure in the sheared Papaku thrust gouges. Because of the low permeability and long diffusion times, which are larger than the duration of the slip pulse, the pore fluid pressure measured at the downstream boundary of the gouge layer is delayed in time with respect to the sample. Therefore, the pressure within the gouge layer needs to be inverted with numerical models before calculating the friction coefficient according to reviewer's request. We did attempt to develop a numerical model of the experiment to estimate pressure and temperature, however we found that constraints on the parameters were poor. For this reason, modelling results are non-unique and we decided to use a different approach. We added an experimental "proof of concept" which is to run the same tests performed on the Papaku gouges on a well-known standard material, Carrara marble.

In the case of the more permeable gouges of Carrara Marble, the diffusion time (0.03-0.14 s) is smaller than the duration of the experiment (0.5 s, see also new Suppl. Fig. 6), and therefore the measured pore pressure at the downstream boundary is representative of the pressure inside the gouge layer. In this case, as suggested by the Reviewer #1, the friction coefficient can be calculated as (Rebuttal Figure 1):

$$\mu = \tau / (\sigma_n - P_{f,d})$$

Rebuttal Figure 1.

This approach allowed us to verify that the friction coefficient of (fluid-pressurized) Carrara marble gouge sheared at 0.8 m/s was similar to the friction coefficient (0.55-0.60) of dry Carrara marble sheared at sub-seismic slip rates (e.g., 0.5-3000 μm/s, Scuderi and Colletini, 2016.). This finding implies that the reduction of shear strength observed in the simulated seismic slip pulses derives from a reduction of effective normal stress by pore fluid pressurization.

Alternatively, we can discuss the role of pore pressure in controlling the shear stress by comparing the measured shear stress vs. slip with the estimated shear stress defined as

$$\tau = \mu(\sigma_n - P_{f,d}).$$

In this latter case we assume a typical value of $\mu=0.55$ for dry Carrara marble gouges (Scuderi and Colletini, 2016) (Rebuttal Figure 2). This figure shows that i) at the onset of slip friction coefficient has to decrease from 1 to 0.55 and ii) the final weakening towards the steady state can be explained by the onset and magnitude of the pressure pulse.

Rebuttal figure 2.

Moreover, by plotting the measured shear stress versus the effective normal stress of the experiment (Rebuttal Figure 3, note the two reference friction coefficient values of 1.0 and

0.55, blue and orange in colour lines, respectively), it is possible to check how at the onset of slip (dark blue data), the decrease in shear stress (from 4.5 MPa to ca. 3 MPa) occurs at constant effective normal stress (i.e., independently of the measured fluid pressurisation). Then, fluid pressurization onsets and we observe the decrease of the shear stress (from ca. 2.75 to 1.2 MPa) with slip. The decrease of the shear stress occurs by reduction of the effective normal stress along the orange line with constant $\mu=0.55$. In the case of Carrara marble, the short delay between initiation of slip and the pressure build up, makes evident that the pressure increase in the slip zone contributes strongly to the low gouge shear strength at slip rates of 0.8 m/s.

Rebuttal figure 3.

To summarize, in the case of Carrara marble the measured pore pressure increase is fast and impacts directly on the shear strength reduction. On contrary, in the clay rich fault material, the pressure increase is slow and it is difficult to trace-back the effect of the pressure pulse on the shear strength without using a model. One relevant difference between the two materials is their permeability that we measured and discussed (see suppl. Fig. 6).

To better explain this important concept, we decided to move the figure of Carrara marble from the Suppl. Material to the main text (now Figure 3) and add the following statements at lines 135-141:

“Carrara marble gouge had a characteristic diffusion time (0.03 s) three orders of magnitude smaller than Pāpaku thrust materials (i.e., 26.9 s, see Methods and Suppl. Fig. 6) and twenty times shorter than the duration of the seismic slip pulse (ca. 0.63 s). Therefore, Carrara marble gouge was drained during the slip pulse and the pressure rise measured by the downstream pressure transducer was detected almost instantaneously to when it occurred in the slipping zone. Conversely, due to their lower permeability, the

pressure increase of the clay-rich gouges of the Pāpaku thrust was detected by the same transducer several tens of seconds after the end of slip²².”

3. As the authors pointed in the manuscript (lines 131–137), there are several mechanisms to increase the pore pressure during the coseismic slip. The authors have enough dataset to distinguish which mechanism can be the dominant for the weakening (shear stress, permeability, porosity change, temperature change, storage capacity, etc). It's not shown in the manuscript but I guess the axial shortening (sample thickness during the experiment, or at least before/ after experiment) is also measured (LVDT is there in Fig. S3a), so they can evaluate the amount of the compaction during the coseismic slip. In my opinion, this kind of analyses is highly required for publishing the research to high impact journals.

We thank the reviewer to point out the relevance of the thickness measurements, which are available for all the experiments presented here. We included in the supplementary materials two additional plots with all the measurements, including the thickness change with slip of the gouge layer (Suppl. Fig. 5).

In this study we wished to discuss (“core message”) the role of fluid pressurisation in dynamic fault weakening, without specifying which process among dehydration, thermal pressurization, and shear fluid compaction were dominant. This because despite knowing the permeability of the materials with good accuracy, other parameters are poorly constrained, like the compressibility of the pores (see comment below), or unknown, like the thermal expansivity of the pores in the clay-rich gouge.

The calculated pore pressure by a thermal pressurization model can be sensitive to the combination of thermal expansivity of the pores (unknown here, but usually 0.0002 1/K, Rice et. al., 2006) and the main parameter controlling the amount of temperature rise: the degree of localization in the deforming layer (unknown here and extremely difficult to estimate without *in-situ* and during shear measurements, an enormous technical challenge). To assess the contribution to the pressure rise of the shear compaction mechanism we should add to the model a feedback between the porosity reduction and the hydraulic diffusivity increase. And we should also consider the reduction in gouge layer thickness which results in the change in position of both upstream and downstream boundary conditions. Moreover, to assess the contribution to fluid pressurization from the dehydration of clays, reaction kinetics should be included, because the short time duration of the experiment would otherwise overestimate the amount of reaction products and therefore the pressure rise (see discussion in Aretusini et al., JGR 2019). But reaction kinetics at our experimental conditions are almost unknown.

Because of the above reasons, though surely interesting, the *ad-hoc* modelling of the relative contribution of dehydration, thermal pressurization and shear enhanced compaction to fault dynamic weakening plus sensitivity analyses of the parameters are beyond the main message of the current work. The main message here is that fluid pressurization occurs and can be detected with dedicated experiments.

4. The pore pressure drop at the initial 0.1 s can be by the fracturing and dilation of the gouge layer. This makes sense to me because fracture energy can be calculated from the decay of the shear stress and the ISCO pump doesn't have capability to keep the pressure constant at such fast change of the pressure.

We agree with the comment of the reviewer: dilation of the gouge layer can be correlated with the achievement of the peak shear strength. So, following the suggestion of the Reviewer (and also of Reviewers #2 and #3) in this revised version, the pressure drops from both downstream and upstream sides for each experiment are included in the Suppl. Tab. 2 and the role of dilatancy discussed at lines 124-127:

“A drop in downstream pore fluid pressure of 0.51 ± 0.11 MPa occurring ca. 0.1 s after the slip initiation was associated to gouge layer thickness increase (dilatancy) of 6 ± 4 μm (Suppl. Table 2). Pore pressure drop and increase of gouge thickness are consistent with shear enhanced dilatancy at slip initiation.”

5. Time delay of the pore pressure recovery can be due to the clogging of the porous plates, may not be due to the low permeability of the gouge. I guess “porous” in the manuscript means just there are several holes on the plates, not porous material like a sandstone.

We addressed this comment in a previous point of this rebuttal.

As the authors described in lines 318–326, 100s after shearing is long enough for re-equilibration of the pore fluid pressure between upstream and downstream lines. Using the same equation, if you calculate the diffusion length of fluid L_c , L_c will be 1 cm after 100 s, which value is more than twice of the sample thickness. So delay of the pore pressure recovery is probably the technical issue, not from the sample.

We agree only in part with the analysis of the Reviewer. In fact, the delay of the arrival of the pore pressure pulse in the experiments performed with the clay-rich gouges from Hikurangi is due to the transport and pore elastic properties of the gouge (not to a “technical” issue as he/she claims) as discussed in the previous points. This is confirmed by the experiments performed with the Carrara marble gouge as discussed before. In the sheared Hikurangi gouges, the delay with respect to slip initiation of the arrival of the peak downstream pressure at the transducer located in the downstream side (see Suppl. Fig. 3) ranges from 19 to 64 seconds (Suppl. Tab. 2). These delay times for the arrival of the maximum downstream pressure are in agreement with the duration, after the initial dilatant stage, of 26 to 108 s for the diffusion of the pressure wave from the centre of the gouge layer to its boundaries (= half thickness of the gouge layer, see Methods). We considered the half thickness of the gouge layer because the propagation of the pressure wave in the upstream half of the gouge layer sample is compensated by the syringe pump. The delay time is indeed related to the gouge transport properties. In fact, the Carrara marble gouge, which has diffusion times ca. 1000 times smaller than the clay-rich gouges from Hikurangi (i.e., 0.03 vs. 26.9 s, respectively), does not show a detectable delay in the detection of the pore pressure build-up. In these permeable gouges, the pressure build up

follows the initial pressure drop associated to gouge dilatancy and the pressure build up detected by the downstream transducer is almost instantaneous to the pressure build up in the slipping zone (Fig. 3). In the revised manuscript, we tried to clarify this concept as follows (lines 135-141):

“Carrara marble gouge had a characteristic diffusion time (0.03 s) three orders of magnitude smaller than Pāpaku thrust materials (i.e., 26.9 s, see Methods and Suppl. Fig. 6) and twenty times shorter than the duration of the seismic slip pulse (ca. 0.63 s). Therefore, Carrara marble gouge was drained during the slip pulse and the pressure rise measured by the downstream pressure transducer was detected almost instantaneously to when it occurred in the slipping zone. Conversely, due to their lower permeability, the pressure increase of the clay-rich gouges of the Pāpaku thrust was detected by the same transducer several tens of seconds after the end of slip²².”

Technically the authors can evaluate the effect of the clogging of the porous plates on the pore pressure change if they run an additional experiment using clayey gouge then measure permeability of the clogged porous plates.

The possible effect of the clogging of the porous plates, which can be ruled out for several reasons as discussed before, is already included in the permeability measurements of these gouges performed with the permeameter (see Suppl. Figure 1). In fact, similar porous plates were used during the permeability measurements (in the permeameter) on both upstream and downstream sides of the gouge layer.

Other comments

I don't think the ISCO syringe pump has enough response for such high speed experiments since the pump can be programmable only up to 1 Hz or even lower. The authors might replace this pump to an air reservoir and air/fluid separator pressurization system for this range of the pressure.

The Reviewer is correct when he states that the logging time of the ISCO pump is 1 Hz. For the experiments presented in this study, the slow response of the pump had the effect of evidencing the dilation in the gouge layer. At the same time, the slow response is not affecting the following stage of the experiment where the fluid pressure increase in the gouge layer was gradually compensated by the syringe pump over hundreds of seconds.

Lines 135–136: thermal dehydration or something equivalent would be better, otherwise add Han et al. (2007) for the case of thermal decarbonation of marble.

We changed “thermal decomposition” to the more correct term “dehydration”.

Some data of pore compressibility is not well solved (e.g., 3rd and 6th row in Table S1). These data must be from the failure of the numerical fitting and not be the real sample data.

We thank reviewer for highlighting this: the minimisation procedure was achieving a wrong minimum. We run another set of optimizations and fixed the wrong values in Suppl. Table 1.

Reviewer #2 (Remarks to the Author):

This is a nice study in which high velocity frictional behaviour of fault gouge recovered from the Hikarangi Trough is tested under fluid saturated, confined conditions. The novelty of the study is not so much in the findings of the study that mirror previous work done, but for the fact that this is the first time that these materials have been confined and tested at seismic slip velocity. Consequently the study verifies and confirms the conclusions drawn in previous studies. I think the authors acknowledge this well and it is good that the results are presented in a balanced way and not oversold. The study will be of wide interest to a range of geoscientists interested in subduction zone seismicity.

I think that the manuscript could do with some polishing. I would suggest that the authors think about fluid pressure diffusion timescales (characteristic times) that would probably show that characteristic times are much longer the duration of the experiments and therefore it is not that surprising that the experiments that they ran give very similar results to the unconfined equivalents ran previously as the mechanical data will be most influenced by the pore pressure built up within the layer which does not have enough time to escape from the layer whether or not the experiment is confined. I think this would be a useful addition and it wouldn't detract from the fact that these are still very technical experiments that have been performed and they do still provide plenty of useful insight into the physics of slip.

I have some specific comments that are detailed below that I hope might improve the clarity of the manuscript in places.

I am happy to clarify with the authors anything that is not clear in my comments.

Dan Faulkner, University of Liverpool, UK.

We thank Dan Faulkner for his very constructive comments and suggestions that allowed us, we hope, to improve significantly the accuracy and the scientific quality of our manuscript. We also thank the Reviewer for acknowledging the novelty of the experimental approach and of the results presented in this manuscript: *“this is the first time that these materials have been confined and tested at seismic slip velocity”*. We did our best to change the manuscript according to his suggestions as shown below.

Line 36. I feel that this may be posed a little more clearly. An important point for me, summarized in the 2011 paper, is that clay-rich forearc sediments should be (are) velocity strengthening and consequently will kill off any ruptures that propagate into this region (this is illustrated by the seismically quiet zone typically seen from the surface to ~10km depth in subduction forearcs).

The authors say that co-seismic weakening will reduce the mechanical work, which is true, but the key factor is the co-seismic weakening results in overall velocity weakening behaviour which is the pre-requisite to the rupture propagating at all. The comment applies to line 47, where I feel it should state that the co-seismic weakening allows for the possibility of rupture propagation and the small energy barrier (presumably from the low breakdown work/fracture energy) will promote large slip.

We agree with the reviewer and reworded into (lines 36-42):

“Theoretical studies suggest that thermal pressurisation of pore fluids trapped in fault materials can reduce the dynamic shear strength of faults during seismic slip¹⁰⁻¹². In the impermeable and velocity strengthening clay-rich materials typical of subduction forearcs, the decrease in fault strength induces a velocity weakening behavior and, since the strength decrease occurs over a short distance, a negligible mechanical work is dissipated by the seismic rupture^{13,14}. The combination of velocity weakening behavior and negligible dissipation of mechanical work makes rupture propagation in shallow sections of the fault possible and also promotes large seismic slip¹⁵.”

Line 65. It would be good for clarity to define what is meant by ‘ductile’ and ‘brittle’ here. These terms are often used loosely and mean different things to different people.

We decided to cut this statement because it added no additional information to the fault structure.

Line 80. This sentence needs modification. While experiments may be a guide to the shear stress evolution during an earthquake, the actual evolution is a highly coupled process that spontaneously evolves according to nature’s boundary conditions, which are certainly not the same in a controlled velocity experiment. Hence the experiments will be a guide to the shear stress evolution, but not equivalent.

The reviewer is correct: changes made.

Line 90. How fast could the upstream pump react? Can it maintain the upstream pore fluid pressure over fractions of a second?

The ISCO syringe pump, has a relatively long response time (see our rebuttal to Reviewer #1) of 1 Hz. Thus it can’t maintain the upstream pore fluid pressure constant at 3 MPa at seismic slip initiation. Despite this apparent limitation, after revising the data set, we are now reasonably convinced that the measured dilatancy is not an artefact but is actually related to a volume increase of the gouge layer partially due to poroelastic properties of the gouge. See also our answer 4 to Reviewer #1. Moreover, gouge dilatancy is not observed during the pre-shearing at 10^{-5} m/s but only when the gouge is abruptly accelerated to 1 m/s. Because of this, the pressure drops from the upstream and downstream sides are now included in Suppl. Tab. 2. The upstream pressure drop is usually occurring within 0.1 s from the start of the seismic slip pulse and its value is of -0.33 ± 0.23 MPa.

Line 91. What was the volume of the downstream reservoir? This is key to report as any

pressure change here will depend on the size of the reservoir. In fact, the ratio of the pore volume of the sample to the downstream reservoir volume will be very useful to estimate.

We measured the volume of the downstream reservoir by using a syringe pump in constant flow mode, observing the variation of the pump volume, to be of 4.67 mL ($4.67 \cdot 10^{-6} \text{ m}^3$). In experiment s1739 before the onset of seismic slip, total volume (calculated from porosity and sample thickness) minus the mass divided by density of the samples gave a pore volume of ca $3.64 \cdot 10^{-6} \text{ m}^3$ in the gouge layer. This gives a rough estimate of the ratio of pore to dead volume as $3.64/4.67 = 0.78$. We included this information at lines (313-314):

“The downstream reservoir had a volume of 4670 mm³, therefore the ratio of pore to downstream reservoir volume was of ca. 0.78.”

Figure 2. These are great results. I am particularly intrigued by the drop in pore fluid pressure as the velocity increases. This is presumably shear enhanced dilatancy (although this depends on how fast acting the pump is, see comment for line 90) which (I think) would be the first time that this has been observed in a HV friction experiment.

We thank and agree with Reviewer for his comment. At first we were cautious of including an extended discussion on dilatancy because of a number of points discussed before. In particular, because there is the possibility that the syringe pump was not fast enough in re-equilibrating the pore fluid pressure. However, we show that the increase of sample thickness occurs indeed during the drop in upstream pore pressure (new Suppl. Fig. 5?) and related comment in lines (124-127). This can be an additional evidence pointing towards the occurrence of shear-enhanced dilatancy.

“A drop in downstream pore fluid pressure of $0.51 \pm 0.11 \text{ MPa}$ occurring ca. 0.1 s after the slip initiation was associated to gouge layer thickness increase (dilatancy) of $6 \pm 4 \text{ }\mu\text{m}$ (Suppl. Table 2). Pore pressure drop and increase of gouge thickness are consistent with shear enhanced dilatancy at slip.”

The immediate recovery of this pressure change is also interesting. The downstream recovers much faster. Why do you think that is? Is it because the slip has localized in the interface with the downstream reservoir? Or is it a pump-related effect?

We think that this effect can be biased by the localization of deformation next to the upstream side of the sample. For identical reasons, we observe in some cases that the ISCO pump appears to be more effective in compensating the pressure drop by shear dilatancy (localization next to rotary side).

Line 104. The longer-term recovery would obviously be much slower in the downstream (uncontrolled) than the upstream. If the response of the upstream pump is immediate (see comment for line 90) then the upstream pressure would sit perfectly at 3MPa.

We agree with reviewer's comment: fluid needs to diffuse through the whole gouge layer thickness to dissipate the downstream reservoir pressure wave. The longer term recover is much slower in fact. After few 100 s, pressure typically equilibrates to 3 MPa on both sides of the gouge layer.

Line 122. I think this comment requires a little more detail before it can be useful to the reader. Do you mean that the experiments on Carrara marble was pretty much drained (i.e. very high permeability), and this is how the pore fluid pressure variations were correlated to the shear stress? If so then it may be worth referencing Faulkner et al. 2018 JGR here, where there are relations on the characteristic timescale for fluid diffusion, given the hydraulic diffusivity. A comparison between the characteristic timescales for the Carrara experiments and the gouge experiments may be useful here (using l^2/k).

Rebuttal Figure 4. (Now Suppl. Fig. 6).

We followed the method presented in Faulkner et al. (2018) to relate the characteristic diffusion time of our sheared gouges to overpressure ΔP and pressurisation rate A. The vertical line in the Rebuttal Figure 4 marks the duration (c. 0.6 s) of our simulated seismic slip pulses. For the less permeable Hikurangi clay-rich gouge and the more permeable Carrara marble calcite-rich gouge, we represent four combinations of half-thickness (0.0012-0.0017 m) and porosity (20-40%), to visualise the upper and lower bounds of their characteristic diffusion times. Thanks to this analysis suggested by the reviewer, it is clearer that the effect of fluid pressure on fault gouges is also controlled by

the time scale of the processes involved. We can't really quantify how big is the control of pressure over shear strength in the case of clays because this attempt would require some modelling to estimate the fluid pressure increase inside the gouge layer as we did for temperature (suppl. Figs 9, 10). Unfortunately, in the case of pressure, modelling results become somewhat *ad-hoc* due to the non-uniqueness of solutions. In any case, we hope that this new graph (new Suppl, Fig 6) will help the reader to follow our reasoning. As a consequence, we reworded the original lines 126-131 and added the reference suggested by the reviewer (lines 134-140):

“Carrara marble gouge had a characteristic diffusion time (0.03 s) three orders of magnitude smaller than Pāpaku thrust materials (i.e., 26.9 s, see Methods and Suppl. Fig. 6) and twenty times shorter than the duration of the seismic slip pulse (ca. 0.63 s). Therefore, Carrara marble gouge was drained during the slip pulse and the pressure rise measured by the downstream pressure transducer was detected almost instantaneously to when it occurred in the slipping zone. Conversely, due to their lower permeability, the pressure increase of the clay-rich gouges of the Pāpaku thrust was detected by the same transducer several tens of seconds after the end of slip²².”

Line 129. I think the terminology could be tighter here. Note that it is the pore fluid pressure that diffuses – not strictly the water. Also diffusion occurs from high pressure to low pressure, so the fluid pressure will be diffusing from the within the layer out into the pore reservoir volumes.

The reviewer is right. Change made. The entire paragraph was reworded, see previous comment

Line 136. This reference needs formatting in the journal style.

Change made.

Line 144. How can vaporization contribute toward pore fluid pressurization? As the temperature increases, the water may start to undergo a phase transformation – this will buffer the temperature – but then won't the water vapour be at the same pressure as the liquid water before the phase transformation? If the water pressure did start to increase would this not buffer the transformation so that more temperature would be required before further transformation can occur?

We agree with the reviewer's excellent remarks. We removed the reference to the vaporization processes controlling fluid pressurization.

Line 144. It is likely that only the smectite will breakdown at these temperatures, but I guess that's enough.

We agree with reviewer's comment, but in reference to poorly bonded water molecules. Therefore, we changed from “decomposition” to “dehydration” to avoid confusion with decarbonation or dehydroxylation.

Line 158. I have the same comment as I had for line 80 – the W_b is not a material property and will depend on the boundary conditions of the experiment and hence it is only a guide to what happens at a propagating rupture tip.

The reviewer is absolutely right. Breakdown work is not a material property (though there still might be some kind of dependence with the mineral composition and the associated coseismic dynamic weakening mechanism) but will depend on normal stress, slip acceleration, etc. We changed the text making clear that W_b calculated from experiments is a guide to what happens in nature during earthquake propagation (see lines 176-177).

“The breakdown work W_b defines the energy dissipated during the loss of fault strength, a quantity that is indicative of the energy subtracted from a propagating earthquake rupture tip^{24–26}.”

The discussion here could also be usefully compared to a recent compilation of w_b values by Seyler et al. 2020 EPSL.

We thank the reviewer for drawing our attention on this work which turned out to be very useful to our discussion. We included a comparison with the collection of W_b in Seyler et al., 2020 (not published at the time of our manuscript submission). However, as discussed in the previous point, a compilation of W_b with other experimental datasets should take into account the differences in: i) whether complete weakening was obtained during our experiments compared to those performed for larger displacements, ii) acceleration and target slip velocity, iii) effective normal stress, and iv) possible role of pre-shearing (and pre-compaction) the gouge layer before the seismic slip velocity pulse (lines 190-194).

“The values of breakdown work of Papaku thrust materials sheared under fluid pressurised and water dampened conditions are comparable with those of clay-bearing materials sampled in the Costa Rica²⁷, Nankai Megasequence²⁸, and Japan Trench^{13,29} sheared at similar seismic slip rate, displacement, and total normal stress³⁰.”

Rebuttal Figure 5. We added our measurements of W_b to the figure 7 from Seyler et al., 2020. Blue circles are clay-bearing wet gouges; black circles are clay-bearing dry gouges. Our values are similar to clay-bearing wet gouges from clay-rich subduction zone samples.

Line 173. This is a problem that all of us have to face – the low normal stress in the experiment is representative of a couple of hundred metres at most – unless overpressured fluids are present.

We agree with the reviewer's comment. This is a common experimental limitation for the rock deformation community performing high velocity friction experiments. It is also possible that overpressured fluids are present in some natural cases.

Reviewer #3 (Remarks to the Author):

This article reports on laboratory high-velocity friction experiments focusing on samples from a splay fault in the Hikurangi subduction zone. In previous high-velocity friction studies, the lack of direct pore pressure measurement was a major drawback, which has been solved here and is an important step which deserves publication.

I do not see anything “wrong” with this study; the data look good and the interpretations are reasonable. I am worried that this study is actually too comprehensive for a short-format journal. Since these are the first measurements using the new technique, they need to be carefully documented. A lot of work has been done for this and this results in several

different “sub-studies”: the comparison between room-dry, wet, and pressurized experiments, the permeability measurements, the comparison between Carrara marble and the Hikurangi samples, and applying the Hikurangi data to the Hikurangi margin itself. I think that both the methods and the data (all the data) deserve to be extensively presented and discussed in a real article and not be tucked away as supplementary material. I certainly understand the desire, or the pressure, to publish in high impact short-format journals, but think that by cramming too much material into a short article that both the data presentation and the discussion suffer.

We thank Reviewer #3 for all her/his intriguing and very constructive comments (e.g, the comment about the shear dilatancy hardening, or the comment about the long lasting post-seismic slip, due to pressurisation of the shear zone). These comments helped us, we hope, to improve substantially the quality of the manuscript. We also thank the Reviewer for pointing out the novelty of our study with respect to previous experimental studies: *“the lack of direct pore pressure measurement was a major drawback, which has been solved here and is an important step which deserves publication”*.

To address also one of the main criticism of the Reviewer, we tried to make the revised manuscript more comprehensive but by keeping this format for Nature Communications. We think that permeability of the tested materials concurs in explaining our mechanical results and the fluid pressurisation via the role of characteristic time of diffusion and should be kept in this article. We discussed in the revised version the role of permeability according to reviewer’s precious suggestions.

Lastly, Reviewers 1 and 2 made in some cases comments similar to those made by Reviewer #3: we will refer our response to their comments in this rebuttal to Reviewer #3.

Other comments:

In Lines, 112-115, a point of emphasis seems to be that the fault core contrasts with the wall rock, but looking at the data in the supplementary material this is a bit misleading, because the behavior of the fault core does not seem very different compared to the wall rocks. For example, it is mentioned that the slip weakening distance is smallest for the fault core, but it is close to the hanging wall values. The residual friction of the fault core is actually higher than that of the wall rocks. The breakdown work is also not very different. I think it is worth acknowledging that properties of the fault core are not much different from the wall rocks, which is consistent with structural observations from the cores, and has some important implications for the distribution of slip at shallow depths.

We agree with reviewer’s point. It is hard to tell but it is possible that the seismic rupture is guided across a particular structural horizon by the slight differences in pore fluid pressure distribution or geometrical properties rather than being controlled by mineral phases and textural properties. Whatever the process active in these type of environments we better explained the possible meaning of our results, adding the reviewer’s suggestion and changing the text as follows (lines 209-210):

“Fault core and wall rocks display similar residual strength and W_b , possibly indicating that seismic slip at shallow depth can propagate in any of these materials.”

I think it is actually important to include the Carrara marble data in the main text. This is because the technique is new, so data needs to be presented on a well-studied standard, and not just on a specific material from a particular field site, which may be unique or unusual.

We agree with Reviewer’s relevant point. Following her/his suggestion, we included the experiment s1823 performed on the more permeable gouges (with respect to the permeability of the Hikurangi ones) of calcitic Carrara marble as a new Figure 3. Please, see also our response to Reviewer #1. In order to include the experiment on Carrara marble, we edited the main text as follows (lines 92-94):

We also performed an additional experiment using higher permeability Carrara marble gouge, which is also a well-studied standard selected to be used as a benchmark.

The article focuses on the pore pressure rise, but I think the pore pressure decrease at the beginning of the experiment is just as important, and something that is rarely discussed. This decrease clearly indicates dilatancy, and looking at Figure 2, it can be seen that by the time the average pore pressure has returned to the original 3 MPa most of the dynamic weakening is already completed. This is important because it seems to argue against thermal pressurization as a cause of dynamic weakening. Also, the dilatancy looks like it drops the pore pressure from 3 to about 2.2 MPa, which is a significant increase in effective stress. Since dynamic weakening occurs anyway, this maybe suggests that dilatant hardening – suggested as a mechanism for SSEs, and of course Hikurangi is famous for SSEs – is not effective. I know that it is popular to discuss pore pressure increases, but in my view the pore pressure decreases are just as important and both need to be discussed.

We agree with these relevant Reviewer’s observations and suggestions. We agree that thermal pressurization alone cannot explain initial dynamic coseismic weakening, and shear-compaction induced pressurisation must also be considered. Similar comments on the role of dilatancy were made also by the Reviewers #1 and #2 and we refer Reviewer #3 to our response to their comments. Because of the suggestions of all the reviewers, we included a new paragraph about the shear gouge dilatancy plus related figures and data:

“A drop in downstream pore fluid pressure of 0.51 ± 0.11 MPa occurring ca. 0.1 s after the slip initiation was associated to gouge layer thickness increase (dilatancy) of 6 ± 4 μm (Suppl. Table 2). Pore pressure drop and increase of gouge thickness are consistent with shear enhanced dilatancy at slip initiation.”

However, we feel that the observations made by Reviewer #3 on Figure 2 might need to consider that the measured downstream pore pressure is delayed in time with respect to the pore pressure at the centre of the gouge layer. The delayed response is due to the fact that Hikurangi gouges are almost impermeable, and diffusion of pore pressure across the

gouge towards the pressure transducer needs time. See response to Reviewers #1 and #2 and the Suppl. Fig. 6 with the characteristic time of diffusion.

Regarding the other relevant comment of the reviewer:

“Since dynamic weakening occurs anyway, this maybe suggests that dilatant hardening – suggested as a mechanism for SSEs, and of course Hikurangi is famous for SSEs – is not effective”.

We would highlight that in the experiments presented here, the imposed loading conditions do not reproduce the natural loading conditions. In fact, we impose a trapezoidal slip velocity function to the sample (see Fig. 2). Clearly, the advantage of this loading history is that it offers relatively simple (with the exception of the acceleration and deceleration stages) relations between shear stress, pore pressure, gouge layer shortening/dilatancy, slip and slip rate. However, the trapezoidal slip velocity function does not consider the elasto-dynamic coupling between stress and slip velocity. Imposing a velocity function to fault patch as we do in the experiments implies the availability of a nominally infinite amount of stress (this will be limited by the torque and power available from the experimental machine) that can be applied on the simulated fault patch. Under these loading conditions, the fault patch will yield, regardless of the initial strain hardening behaviour. Now, if SSE are thought to nucleate in the drilled fault patch, in the presence of a natural and finite shear stress acting on the fault patch, the shear dilatancy hardening observed in this study could be large enough to overcome the local shear stress, arresting slip. So we believe that these experimental observations are still in agreement with a fault gouge material that may produce SSE because the energy balance should be always included in the process of extrapolation.

Moving on to the pore pressure increase, it seems like an important factor here is that the pore pressure increase is long-lasting, much longer lasting than the slip pulse here, and by extension probably much longer than a (main) fast slip event in the Hikurangi. It seems to be important because it implies that the fault is weakened for an extended period of time following a slip event, but it is not discussed.

We did not think about this other relevant implication of the experimental results presented in this study made by the Reviewer, and we thank her/him for the input. To address it, we added the following paragraph in the discussion (lines 210-214):

“Moreover, it is possible that, due to the low permeability of shallow sediments in the surrounding of the Papaku splay thrust, the pore fluid pressure wave developed during seismic slip remains trapped in the fault core. This may result in a prolonged weakening of the fault and might affect the magnitude of initial afterslip and the occurrence of post-seismic creep or SSEs after the earthquake mainshock.”

Which brings me to my final main comment, that these experimental results aren't very well connected to the Hikurangi subduction zone and to the specific behaviors that occur there. As a side note, I have to point out that it is unknown if SSEs or tsunami earthquakes actually occurred on the fault drilled here. But anyway, the conclusion here is that thermal

pressurization can contribute to the propagation of earthquakes or tsunami earthquakes to shallow depths. Ok, in general that is true. But does this apply to SSEs? Ordinary earthquakes? Tsunami earthquakes? All three?

We apologize with reviewer for our lack of clarity about this point. We wanted to apply the results to tsunami earthquakes as there were two tsunamis occurring in the area in 1947. This is now specified in the main text at lines 203-206:

Indeed, at the Hikurangi subduction zone, the IODP Expedition 375 drilled the shallowest part of the Pāpaku thrust branching off the plate decollement above the zone where SSE are recorded⁵ and near the rupture area of the two 1947 tsunami earthquakes³.

Moreover, as discussed in a previous reply to Reviewer #3, the shear dilatancy hardening behaviour of these gouges may imply that the gouges recovered from the Papaku thrust, depending on the loading conditions, may also accommodate SSEs.

And it is stated that W_b and D_w are small. This needs to be qualified somehow - compared to what? Marble? Or other subduction zones? A good approach could be to compare these measured values with the comparison published recently by Seyler et al. (2020 EPSL).

We thank the reviewer to point out this remarkable work. A very similar comment was made by Reviewer #2. Please, refer to our comment/rebuttal (lines 190-194).

“The values of breakdown work of Papaku thrust materials sheared under fluid pressurised and water dampened conditions are comparable with those of clay-bearing materials sampled in the Costa Rica²⁷, Nankai Megaseplay²⁸, and Japan Trench^{13,29} sheared at similar seismic slip rate, displacement, and total normal stress³⁰.”

Line 10: I suggest shortening the text here and not referring to the Tohoku earthquake, because the reader might mistakenly think the paper is about Tohoku. It's fine to have this in the introduction, just not in the abstract.

We agree with the reviewer and removed the following phrase from the abstract:

“In the case of the tsunamigenic 2011 Mw 9.0 Tohoku-Oki earthquake, up to 50 m of fault slip was accommodated at shallow depth.”

Line 41: “non-lithified” is better than “non-cohesive” here

Change made.

Line 60-61: I don't think it is known that the Papaku fault has hosted any kind of tsunami earthquake or SSE, especially at the drill site. Or do you mean the plate boundary fault? Make this clearer.

We clarified that the plate boundary fault hosted tsunamis and SSE (see line 63-64).

“In this area, the Pāpaku thrust is a shallow branch of the plate boundary fault which has hosted historic tsunami earthquakes and more recently, shallow SSEs (Figs. 1a-b).”

Line 73: are the permeability measurements done on intact samples? How were they oriented relative to the flow direction?

Thank you for the clarification: permeability was performed on remoulded samples (now it's specified in the main text. The samples are oriented so that flow direction is perpendicular to the gouge layer base. This now specified in the methods at lines 247-248 as follows:

“Because of the sample assemblage, flow direction was perpendicular to the gouge layer base.”

Line 119-120: what was the maximum decrease in pore pressure?

We specified the amount of decrease in pore fluid pressure and the increase in gouge layer thickness at slip initiation and added these values in the Supplementary Table 2.

Line 194: Figure 4b is confusing to me. First, it would be helpful to distinguish better the rupture velocity and the slip velocity.

Change made.

Second, behind the rupture front is denoted as “slipping” but at what velocity? if V has dropped to zero, it shouldn't be slipping.

The Reviewer is right: we changed the label so now the fault trace is indicated.

Is the shear stress then low because it is at its dynamic value? Or is it because the fault is unloaded?

According to our experimental evidence, the shear strength decreases down to its dynamic value.

To the far left, the pore pressure pulse has worn off, but if the fault is slipping shouldn't the pressure still be elevated?

We agree with Reviewer's remarks also in this case. Change made.

Updated figure 5.

REVIEWER COMMENTS

Reviewer #1 (Remarks to the Author):

Now I read the revised manuscript and the rebuttals by Aretusini et al. I think the authors answered most of reviewers' comments. Please find my comments below.

Rempe et al., (2017, JSG; 2020, JGR) conducted high velocity friction experiments on Carrara marble gouge under a pore pressure control (as much as I know, Rempe et al., 2017 is the first reports which conducted HVF experiments under the Pp control). Actually, while experiments by Rempe doesn't show any pressure drops using same sample at almost same condition or even high pressure conditions, they reported the similar dynamic weakening. The authors need to address how their work is similar to or different from the present study. Somehow these papers are not referred in the manuscript while some names of coauthors in the ms is also in Rempe's papers. It's somewhat insincere for the first author.

Accumulation of our knowledge (of course this manuscript contributes to that) on high velocity friction experiments of various materials revealed that everything gets weak at high speed by multiple weakening mechanisms (e.g., Di Toro et al., 2011). Basically, the authors are trying to demonstrate that one of weakening mechanisms is important for Hikurangi subduction zone from only few special experiments under very limited experimental condition. While they used a "precious sample" from a special place, the authors look like too hastily to conclude that. I understand the importance of their work, just I think further experimental investigations are required for this study and/or in future study.

Reviewer #2 (Remarks to the Author):

Review of revised manuscript "Fluid pressurisation and earthquake propagation in the Hikurangi subduction zone" by Aretusini et al.

My opinion of this manuscript has not changed greatly since my first review. I still feel that the experimental results confirm what has been previously proposed but that the real novelty comes from the step forward in the experimental set up that was used where the pore fluid was confined and the pressure measured during the slip history of the gouge. It was instructive to see the comments of the other reviewers and I have some sympathy with the view of reviewer 3 in that the methods and results may be better presented all together in a longer format journal. However, the authors have heavily utilised the supplementary materials and this does complete the description of the study. The question remains for the authors on whether they want the focus of the manuscript to be on the main findings from the study that, as noted above, confirm what has been previously proposed, or whether they want to shift that focus to include the description, novelty and analysis of the experiments conducted. I guess that the submission/resubmission of the manuscript answers that question and I am certainly happy to support the publication in Nature Communications despite the fact that I am not sure that quite as many people will read the content in supplementary materials as perhaps they might if this work was published all together in a longer format.

In regard to the response to my comments, the authors have done an excellent job. The additions tighten up the findings of the manuscript considerably and the authors responded thoughtfully and carefully to all the points I raised. I think the characteristic timescales figure adds to the understanding of the Carrara and Hikurangi experiments. Perhaps providing a key to the deltaP curves might make the figure slightly more visually appealing than arrows to each of the lines? Overall, I am

satisfied that this noteworthy study is fully documented/justified, and tells a nice story using some very novel equipment developments.

Dan Faulkner, Liverpool, January 2021

Reviewer #3 (Remarks to the Author):

This revised manuscript is improved compared to the original. There are some things that I still feel need to be clarified, and some of the original review comments should be addressed more thoroughly. That said, with some improvement this could be a very impactful paper.

There are some things about the pore pressure control/measurement that are unclear to me. It looks like the pore pressure measurement was different for the Carrara marble and the Hikurangi materials – for the Hikurangi experiments the pressure is measured at both ends (undrained boundary condition), but for Carrara it is imposed to be 3 MPa at one end and it is measured at the other end (one drained, one undrained boundary) (Lines 85-86 and 95-97). Is this the case, and if so, why the difference? And how was the background fluid pressure of 3 MPa then imposed during the Hikurangi experiments – was it brought to 3 MPa in pressure control, then switched to measurement mode before shearing? In Fig. 3, the pressures are constant before and after the slip pulse, as if they are maintained in pressure control, whereas the text suggests that they are both set to measure, not apply, the pore pressure.

It is reported that the changes in pore pressure reported for the Carrara marble are larger than those for the Hikurangi samples (Lines 124-129 and 131-139), but I do not see how the Carrara sample can be drained if you observe such large pressure fluctuations. Do you mean the Carrara marble is expected to be well-drained due to high permeability, and you observed large pressure changes due to the imposed undrained boundary conditions at both ends?

If, contrary to my understanding, the pore pressure measurement/control was exactly the same for both the Hikurangi and Carrara experiments, then the pressure differences are fairly easily explained by the larger dilatancy in the Carrara marble. This makes for an interesting direct comparison which highlights the hydromechanical behavior of two very different materials during fast slip, which deserves to be highlighted (of course assuming that the experiment boundary conditions are the same and can actually be compared).

Finally, as noted in the comments on the first manuscript, the dilatancy effect and pore pressure reduction is important, not just the pore pressure increase. I understand the authors' argument that the sample is forced to reach coseismic slip rates so that the effectiveness of dilatancy hardening in stopping or slowing a slip event can't really be evaluated. But, the authors have hard evidence that dilatancy does occur early in the slip event, which causes substantial pore pressure decrease (and thus normal stress and shear stress increase). I think this is a key observation that needs to be highlighted more prominently (e.g. by adding this to the abstract and conclusions). Most studies focus on pore pressure increases and weakening, so including discussion of the dilatancy observation will make the paper stand out compared to previous work and ultimately increase the impact of the paper.

Lines 38-41: I don't think this should be called "velocity weakening". Yes, the material weakens as it accelerates, but the experimental boundary conditions and weakening mechanisms are different compared to the rate-and-state velocity-weakening friction, which is measured under controlled changes in slip velocity and with the friction at steady state before and after the step. "Dynamic

weakening" is more appropriate.

Line 78: I assume you report the values from 3 MPa, because it approximates the in-situ condition of the borehole samples, right? This is worth mentioning. Reading further I see this is mentioned at Lines 195-198, but it should come here, or somewhere early in the text when describing the experiments.

Lines 100-102: May be relevant to mention that between the slip pulses, the shear stress is brought to zero, so it is initially fully unloaded before the high-velocity pulse (right?)

Lines 118-121: I think it is still important to mention here that the fault core (under fluid pressurized conditions) is not significantly weaker than the footwall or hanging wall. This sets it apart from other fault zones where the fault core is much weaker and represents a large contrast with the wall rock – e.g. at Tohoku and SAFOD.

Lines 190-194: I would like to see a bit more rigorous comparison than just saying that the values are "comparable". At least, give the values for the other subduction zones, and/or plot them along with the Hikurangi values from this study (e.g. in Fig 4c somehow).

Fig. 5b: the pore pressure curve in this schematic is a bit misleading, because it does not really correspond to the data - it shows a very small pressure drop followed by a large pressure increase, whereas the experiments show that the pore pressure drop is at least as large, if not larger than the following pore pressure increase.

REVIEWER COMMENTS

Reviewer #1 (Remarks to the Author):

Now I read the revised manuscript and the rebuttals by Aretusini et al. I think the authors answered most of reviewers' comments. Please find my comments below.

We thank the reviewer for their appreciation of the efforts we made to address their relevant points.

Rempe et al., (2017, JSG; 2020, JGR) conducted high velocity friction experiments on Carrara marble gouge under a pore pressure control (as much as I know, Rempe et al., 2017 is the first reports which conducted HVF experiments under the Pp control). Actually, while experiments by Rempe doesn't show any pressure drops using same sample at almost same condition or even high pressure conditions, they reported the similar dynamic weakening. The authors need to address how their work is similar to or different from the present study. Somehow these papers are not referred in the manuscript while some names of coauthors in the ms is also in Rempe's papers. It's somewhat insincere for the first author.

The reviewer is correct: the experiments described in the paper published by Rempe et al. in the Journal of Structural Geology (2017) and in Rempe et al, Journal of Geophysical Research (2020) were performed on calcite gouges (1) without Pp control with SHIVA in the HP-HT INGV labs in Rome and, (2) with Pp control with the Pressurized High-Velocity apparatus (PHV) installed at the Kochi Institute for Core Sample Research/JAMSTEC in Nankoku, Japan.

In the first submitted version of our manuscript we did not quote these previous papers by Rempe et al., because the experiments discussed in the main text were performed on Hikurangi clay-rich gouges and calcite gouge experiments were added only after the first review round. Moreover, the paper Rempe et al., 2020 was accepted for publication at the end of 2020, so during the revision of our manuscript. The current revised version submitted to your attention quotes both these papers by Rempe et al.

B

Reb_Fig.1 (A) Sketch of the gouge holder installed in the PHV machine at JAMSTEC (from Rempe et al., JGR 2020). Fluid will pass through the tubes labeled In and Out, and through the sample chamber and the gouge layer that is marked in green and has an outer diameter of 60 mm. Gouge layer is not loaded in the sketch. (B) Sketch of the gouge holder used in the experiments discussed in this manuscript (Suppl. Fig. 3).

In the revised version, we added a comment concerning the similarities and differences between our experiments and those performed by Rempe et al., (2017, 2020, LL. 164-171). The main differences are related to the experimental configuration (solid- versus fluid- confinement, location of the pressure transducers, etc.) and the type of gouge used (see Reb_Fig.1):

- 1) In the PHV machine, because of the solid-solid (= solid confinement) interface between the Teflon liners and the upper and lower holder walls, the water-pressurized clay-rich gouges could escape from the slip zone. Note that gouge escape does not happen in the same vessel for water-pressurized calcite gouges because of the different rheology, Poisson's ratio of the material, and larger grain size. Instead, due to the experimental assembly presented in this study under review, fault gouges were laterally confined by a pressurized heat shrink membrane (= fluid confinement), so that spurious gouge layer extrusion was very limited or absent.
- 2) In the PHV machine, pore pressure was measured at large distance from the gouge (at least 180 mm, outside of the sketched area of the Reb. Fig. 1) combined with a large water storage volume relative to the gouge layer (measured from technical drawing to be at least ca. 25.5 mL) which buffers pore pressure changes (Reb. Fig. 1a, and discussion in Brantut and Aben 2021 GJI). Instead, in the new experimental configuration used here (Reb. Fig. 1b), the pressure transducers were positioned close to the gouge layer (i.e., a small upstream transducer positioned ~3 mm from the slip zone, and a downstream transducer ~70 mm away) combined with a relatively small downstream reservoir volume (measured with our ISCO pump to be ~4.7 mL), which combines to result in a better dynamic sensor response in comparison with Rempe et al. 2020.

In fact, the issues identified above motivated our efforts to develop the experimental assembly presented in this manuscript.

Accumulation of our knowledge (of course this manuscript contributes to that) on high velocity friction experiments of various materials revealed that everything gets weak at high speed by multiple weakening mechanisms (e.g., Di Toro et al., 2011). Basically, the authors are trying to demonstrate that one of weakening mechanisms is important for Hikurangi subduction zone from only few special experiments under very limited experimental condition. While they used a “precious sample” from a special place, the authors look like too hastily to conclude that. I understand the importance of their work, just I think further experimental investigations are required for this study and/or in future study.

We agree with the reviewer and acknowledge in the conclusions of the revised version that further experimental investigations are required given the heterogeneity of the mineral composition of the subducted sediments in Hikurangi and other active margins: see LL. 248-251.

Reviewer #2 (Remarks to the Author):

Review of revised manuscript “Fluid pressurisation and earthquake propagation in the Hikurangi subduction zone” by Aretusini et al.

My opinion of this manuscript has not changed greatly since my first review. I still feel that the experimental results confirm what has been previously proposed but that the real novelty comes from the step forward in the experimental set up that was used where the pore fluid was confined and the pressure measured during the slip history of the gouge. It was instructive to see the comments of the other reviewers and I have some sympathy with the view of reviewer 3 in that the methods and results may be better presented all together in a longer format journal. However, the authors have heavily utilised the supplementary materials and this does complete the description of the study. The question remains for the authors on whether they want the focus of the manuscript to be on the main findings from the study that, as noted above, confirm what has been previously proposed, or whether they want to shift that focus to include the description, novelty and analysis of the experiments conducted. I guess that the submission/resubmission of the manuscript answers that question and I am certainly happy to support the publication in Nature Communications despite the fact that I am not sure that quite as many people will read the content in supplementary materials as perhaps they might if this work was published all together in a longer format.

We thank the reviewer for his insights. Following his comments, we tried to shift the focus to consider the novelty of our experiments and the role of gouge layer dilatancy induced strength changes during high velocity sliding, as also suggested by reviewer 3 (see LL 159-174). Moreover, we plan to further explore investigate the role of dilatancy during high velocity slip in future studies.

To improve accessibility of the Suppl. Materials we merged Old Fig. 2 and 3 into New Fig. 2, in which we moved the thickness measurements (which was in Suppl. Fig. 5), we also moved Suppl. Fig. 6 (diffusion times) into the main text as New Fig. 3.

In regard to the response to my comments, the authors have done an excellent job. The additions tighten up the findings of the manuscript considerably and the authors responded

thoughtfully and carefully to all the points I raised. I think the characteristic timescales figure adds to the understanding of the Carrara and Hikurangi experiments. Perhaps providing a key to the ΔP curves might make the figure slightly more visually appealing than arrows to each of the lines? Overall, I am satisfied that this noteworthy study is fully documented, justified, and tells a nice story using some very novel equipment developments.

We sincerely thank the reviewer for his comments. We included a legend in the supplementary figure 6 (now New Fig. 3) hopefully making the figure more aesthetically pleasing whilst improving the clarity of our message.

Dan Faulkner, Liverpool, January 2021

Reviewer #3 (Remarks to the Author):

This revised manuscript is improved compared to the original. There are some things that I still feel need to be clarified, and some of the original review comments should be addressed more thoroughly. That said, with some improvement this could be a very impactful paper.

We thank the reviewer for their insightful comments. Their comments regarding the role of gouge dilatancy in controlling the evolution of fault strength during simulated seismic slip pulses are already motivating our experimental studies.

There are some things about the pore pressure control/measurement that are unclear to me. It looks like the pore pressure measurement was different for the Carrara marble and the Hikurangi materials – for the Hikurangi experiments the pressure is measured at both ends (undrained boundary condition), but for Carrara it is imposed to be 3 MPa at one end and it is measured at the other end (one drained, one undrained boundary) (Lines 85-86 and 95-97). Is this the case, and if so, why the difference? And how was the background fluid pressure of 3 MPa then imposed during the Hikurangi experiments – was it brought to 3 MPa in pressure control, then switched to measurement mode before shearing? In Fig. 3, the pressures are constant before and after the slip pulse, as if they are maintained in pressure control, whereas the text suggests that they are both set to measure, not apply, the pore pressure.

We are sorry for the misunderstanding we caused for reviewer #3. In our tests we continuously monitor the pressure transducers independently from the control of the pore fluid pressure (= 3 MPa) which is performed separately by the ISCO pumps. These are constant fluid pressure experiments and we measure local variations in pore pressure close to the gouge layer.

The same identical procedure and boundary conditions were used for Carrara marble gouge and Papaku thrust materials: the pore fluid pressure in the gouge layer is imposed at a constant value of 3 MPa on the upstream side of the gouge layer, by keeping the valve to the left of the pore pressure Isco pump open (Fig 3 suppl mat. or Reb. Fig. 1b).

We partly reworded the original text to clarify better the boundary conditions of the experiments in LL. 95-99 (see also Methods, LL. 363-364).

It is reported that the changes in pore pressure reported for the Carrara marble are larger than those for the Hikurangi samples (Lines 124-129 and 131-139), but I do not see how the Carrara sample can be drained if you observe such large pressure fluctuations. Do you mean the Carrara marble is expected to be well-drained due to high permeability, and you observed large pressure changes due to the imposed undrained boundary conditions at both ends?

Carrara fault gouge is expected to be well-drained (New Fig. 3, diffusion times) but, because of gouge dilatancy and compaction during simulated seismic slip (evidenced by the LVDT measurements, New Fig. 2), large pressure changes in the slip zone occur. There pore pressure variations are measured by the transducers located upstream and downstream of the gouge layer (Reb. Fig. 1b).

If, contrary to my understanding, the pore pressure measurement/control was exactly the same for both the Hikurangi and Carrara experiments, then the pressure differences are fairly easily explained by the larger dilatancy in the Carrara marble. This makes for an interesting direct comparison which highlights the hydromechanical behavior of two very different materials during fast slip, which deserves to be highlighted (of course assuming that the experiment boundary conditions are the same and can actually be compared).

We confirm that the boundary conditions of the fluid pressurised experiments are the same. The experimental results very much agree with the reviewer's interpretation that the larger drop in pore pressure for Carrara marble gouge is due to the difference in dilatancy because of the different hydromechanical behavior of the two materials. However, we did not stress much this point in the main text for two main reasons:

- 1) the lower permeability of Hikurangi gouges acts to filter out dynamic pore pressure changes, making direct interpretation of our data challenging: pore pressure drops or increase are time delayed with respect to dilatancy or compaction, respectively.
- 2) to assess the occurrence of gouge dilatancy, a constant volume condition on the upstream side has to be ensured (as stated before, in our pressurised experiments we used constant pressure condition). Therefore, in the future, more experiments under constant volume conditions are needed.

Here, we chose to work under constant pressure conditions. This was before we realized the occurrence of dilatancy as the reviewer evidenced during the review process. In the future more experiments will be performed under constant volume conditions with the aim to investigate changes the physical relations between dilatancy and pore pressure drop (we added this point to the conclusions in LL 248-249).

Finally, as noted in the comments on the first manuscript, the dilatancy effect and pore pressure reduction is important, not just the pore pressure increase. I understand the authors' argument that the sample is forced to reach coseismic slip rates so that the effectiveness of dilatancy hardening in stopping or slowing a slip event can't really be evaluated. But, the authors have hard evidence that dilatancy does occur early in the slip event, which causes substantial pore pressure decrease (and thus normal stress and shear stress increase). I think this is a key observation that needs to be highlighted more prominently (e.g. by adding this to the abstract and conclusions). Most studies focus on pore pressure increases and weakening, so including discussion of the dilatancy

observation will make the paper stand out compared to previous work and ultimately increase the impact of the paper.

We thank the reviewer. The main motivation why we were initially skeptical about dilatancy is exactly outlined by the reviewer (and in our previous comment about the necessity of constant volume boundary conditions).

However, due to reviewer #3's point, we made the decision to change the main text to highlight the role of dilatancy in the abstract (LL 15-16), discussion (adding the new paragraph in LL 159-174), and conclusions (LL. 245-248). This point could also address the point about novelty of this study suggested by rev. 2.

Lines 38-41: I don't think this should be called "velocity weakening". Yes, the material weakens as it accelerates, but the experimental boundary conditions and weakening mechanisms are different compared to the rate-and-state velocity-weakening friction, which is measured under controlled changes in slip velocity and with the friction at steady state before and after the step. "Dynamic weakening" is more appropriate.

We agree with reviewer #3's point. Changes made. This point addressed a previous comment of reviewer #2, and we tried to reconcile the two views expressed by reviewer #3 and #3 in the current version (LL 36-40).

Line 78: I assume you report the values from 3 MPa, because it approximates the in-situ condition of the borehole samples, right? This is worth mentioning. Reading further I see this is mentioned at Lines 195-198, but it should come here, or somewhere early in the text when describing the experiments.

We agree with reviewer #3's point. Changes made (LL 79-80).

Lines 100-102: May be relevant to mention that between the slip pulses, the shear stress is brought to zero, so it is initially fully unloaded before the high-velocity pulse (right?)

We agree with reviewer #3's point. Changes made (LL 357-358).

Lines 118-121: I think it is still important to mention here that the fault core (under fluid pressurized conditions) is not significantly weaker than the footwall or hanging wall. This sets it apart from other fault zones where the fault core is much weaker and represents a large contrast with the wall rock – e.g. at Tohoku and SAFOD.

We agree with reviewer #3's point. This loss of contrast may result from the loss of fault microstructure due to remolding of the samples, which is a necessity for our sample geometry. Changes made (LL 128-129).

Lines 190-194: I would like to see a bit more rigorous comparison than just saying that the values are "comparable". At least, give the values for the other subduction zones, and/or plot them along with the Hikurangi values from this study (e.g. in Fig 4c somehow).

We agree with reviewer's comment. We changed Figure 4c so that now our data are compared with breakdown work and slip weakening distance data from Seyler et al., 2020 (and references therein). In particular, we compared our results with the those obtained from experiments performed on fault gouges with phyllosilicate content similar to Hikurangi

materials (i.e., 45-55 wt.% phyllosilicate content).

Fig. 5b: the pore pressure curve in this schematic is a bit misleading, because it does not really correspond to the data - it shows a very small pressure drop followed by a large pressure increase, whereas the experiments show that the pore pressure drop is at least as large, if not larger than the following pore pressure increase.

We agree with reviewer's comment. We changed Figure 5b so that now the pressure drop by shear-induced dilatancy is of similar magnitude to the pressure increase resulting from compaction (and other thermal effects).

REVIEWERS' COMMENTS

Reviewer #1 (Remarks to the Author):

I read the revised manuscript and now I think the manuscript can be acceptable to the journal. Rempe et al., (2017) is missing in the reference list, and Seyler et al. (2020, EPSL) and Rempe et al. (2020, JGR) are listed twice. There are many garbles and missing information in the reference list as well. Please final check the reference list.

Reviewer #3 (Remarks to the Author):

The authors have done a good job addressing my review comments. I am satisfied with the changes and responses, and support publication.

REVIEWERS' COMMENTS

Reviewer #1 (Remarks to the Author):

I read the revised manuscript and now I think the manuscript can be acceptable to the journal. Rempe et al., (2017) is missing in the reference list, and Seyler et al. (2020, EPSL) and Rempe et al. (2020, JGR) are listed twice. There are many garbles and missing information in the reference list as well. Please final check the reference list.

We checked the reference list and corrected it.

Reviewer #3 (Remarks to the Author):

The authors have done a good job addressing my review comments. I am satisfied with the changes and responses, and support publication.

We thank the reviewer for their comment.